# Magnetotactic bacteria optimally navigate natural pore networks

**Alexander P Petroff\*, Julia Hernandez, Vladislav Kelin, Nina Radchenko-Hannafin**

Department of Physics, Clark University, Worcester Massachusetts, United States

## eLife Assessment

Combining experiments in microfluidic devices and computer simulation, this study provides a **valuable** analysis of the relevant parameters that determine the motility of (multicellular) magnetotactic bacteria in sediment-like environments. The study presents **convincing** evidence that there is an optimum in the biological parameters for motile life under such conditions.

**\*For correspondence:**
apetroff@clarku.edu

**Competing interest:** The authors declare that no competing interests exist.

**Abstract** Magnetotactic bacteria swim along geomagnetic field lines to navigate the pore spaces of water-saturated sediment. To understand the physical basis for efficient navigation in confined geometries, we observe the motion of multicellular magnetotactic bacteria through an artificial pore space under an applied magnetic field. Magnetotaxis is fastest when bacteria swim a distance that is of order the pore size in the time required to align with the applied field. A model—in which bacteria deterministically align with the magnetic field and randomly scatter off boundaries—predicts the observed non-monotonic relationship between the drift velocity and applied magnetic field and the value of the maximum drift velocity. A comparison of the reported values of the magnetic moments, swimming speeds, and hydrodynamic mobilities across diverse magnetotactic bacteria reveals that these variables covary such that the average speed of magnetotaxis of each species is close to optimal for its natural environment.

## Introduction

Magnetotactic bacteria represent a phylogenetically diverse group—including species from Desulfobacterota, several branches of Proteobacteria, and Nitrospira—that use the geomagnetic field to navigate water-saturated sediment (*Lefèvre et al., 2014*; *Delong et al., 1993*). The dynamics of magnetotaxis are similar across all known bacterial examples (*Klumpp et al., 2019*; *Frankel et al., 2007*). A cell, or consortium of cells (*Abreu et al., 2007*; *Schaible et al., 2024*), uses flagella to exert a force that is parallel (*Frankel, 1984*; *Esquivel and De Barros, 1986*) to its permanent magnetic moment, which is produced by biologically precipitated inclusions of magnetite or greigite called magnetosomes (*Faivre and Schüler, 2008*). The magnetic torque on a bacterium turns it like a compass needle to align its magnetic moment with magnetic field lines. This motion is purely physical and requires no behavioral adaptations by the organism (*Frankel, 1984*; *Esquivel and De Barros, 1986*). As magnetotactic bacteria swim parallel to their magnetic moments, they move along field lines. Because the geomagnetic field is rarely parallel to the sediment surface, magnetotaxis moves bacteria vertically through the sediment, along the dominant chemical gradients that shape microbial life in the top several centimeters of sediment (*Blakemore and Frankel, 1981*; *Lefèvre et al., 2014*). This biased motion allows magnetotactic bacteria to position themselves in changing chemical gradients (*Bazylinski and Frankel, 2004*; *Smith et al., 2006*) and to shuttle between vertically stratified chemical environments (*Li et al., 2020*).

**eLife digest** Animals have developed a variety of strategies to detect direction, from specialised cells and organs to complex brain structures. This can be particularly difficult for microorganisms, such as bacteria, which often live in microenvironments where gravity cues are generally negligible.

For example, magnetotactic bacteria – bacteria that sense and orient along magnetic fields – live in waterlogged environments and instead use the geomagnetic field, the Earth's magnetic field, to navigate. They contain magnetic crystals that act like a compass needle, aligning the cells with magnetic field lines. Because geomagnetic field lines are rarely parallel to the sediment surface, these bacteria move vertically through sediments. This motion allows them to position themselves at precise depths, follow changing gradients, and move between layers with distinct chemical conditions.

However, moving vertically through the sediment is challenging. First, a microbe must couple its motion to some external field with up-down asymmetry. Several options exist, including the gravitational field and gradients of chemicals or light. It must then balance motion along field lines with the need to move around obstacles.

Despite their similar behaviors, magnetotactic bacteria show a large diversity in their morphological and motility traits. Petroff et al. sought to understand how natural selection has shaped these traits to balance vertical motion with efficient obstacle avoidance.

Using a combination of theory and experiment, the researchers explored how swimming speed, magnetic moment and size of an organism affect navigation through porous environments. They measured the average speed of the magnetotactic bacterium *Magnetoglobus* through an artificial pore space in a controlled magnetic field.

The results showed that efficient navigation requires an optimal combination of swimming speed, magnetic moment and body size. Movement along magnetic lines was fastest when the distance a microbe swims before realigning with the magnetic field matches the pore size. Microbes that reorient too quickly get trapped; those that align too slowly wander. Additionally, because the efficiency of magnetotaxis only depends on the ratio of environmental factors and bacterial phenotype, there is no single characteristic that can be tuned to allow optimal navigation. Rather, optimality is achieved when different phenotypic characteristics balance one another. For example, small cells with weak magnetic moments perform as well as large cells with strong ones.

Magnetotactic bacteria have found a remarkable solution to navigating complex environments. Natural selection has shaped their phenotypes so that the environmental randomness is mirrored in their trajectories. Similar strategies may also apply to artificial navigation in complex settings, such as robots designed to move through rubble. To this end, it could be useful to design magnetotactic robots.

Despite the similarity of their motion, a comparison of different species reveals tremendous diversity (*Lefèvre et al., 2014*). Magnetotactic bacteria may be cocoidal (*Keim et al., 2007*; *Acosta-Avalos et al., 2019*), rod-like (*Spring et al., 1993*), or spiral (*Bazylinski and Frankel, 2004*) with lengths between 1 μm and 20 μm (*Esquivel and De Barros, 1986*). While most species are unicellular, multicellular consortia are also common (*Lefèvre et al., 2014*). Swimming speeds vary by an order of magnitude, from 12 μm/s (*Esquivel and De Barros, 1986*) to more than 140 μm/s (*Bente et al., 2020*). Magnetic moments vary by two orders of magnitude, from $\approx 0.5$ fAm$^2$ to 54 fAm$^2$ (*Esquivel and De Barros, 1986*). A table of the typical phenotypic characteristics of diverse magnetotactic bacteria is available in Appendix 1.

Here, we investigate how magnetotactic bacteria tune their phenotypes in order to balance motion along magnetic field lines with the necessity to move around obstacles such as sand grains. We have previously argued (*Petroff et al., 2022*) that there is an optimal combination of magnetic moment, cell size, and swimming speed that maximizes the average speed of magnetotactic bacteria through a pore space. This hypothesis was recently independently proposed and tested by *Codutti et al., 2024*, who found that the average speed of magnetotactic bacteria through an artificial pore space is indeed maximized if the magnetic torque on a bacterium is tuned. This analysis is closely related to the work of *Dehkharghani et al., 2023*, who examined the effect of obstacle geometry on motion of magnetotactic bacteria through a complex pore space. In this paper, we first show that the speed of

a specific type of magnetotactic bacteria through an artificial pore space is maximized if the distance it swims while aligning with the ambient magnetic field is tuned to the pore size. Proceeding from a literature review, we then argue that a variety of unrelated magnetotactic bacteria from around the world are all optimally efficient navigators of their native pore spaces.

We choose to study members of the Desulfobacterota genus *Magnetoglobus*, which are generally referred to as *multicellular magnetotactic bacteria* (MMB) (*Abreu et al., 2007*). MMB are the only type of bacteria that are known to lack a unicellular stage in their life cycle (*Keim et al., 2004*). Groups of several tens of non-clonal cells (*Schaible et al., 2024*) are physically attached together and coordinate their motion and growth to behave like a single multicellular organism, which is called a *consortium*. Individual bacteria are not observed to join or leave existing consortia. Rather, new consortia form when existing consortia divide symmetrically (*Keim et al., 2004*). MMB consortia are spherical with a typical diameter $a = 3 - 8$ μm and composed of a monolayer of cells (*Leão et al., 2017*). Each cell in a consortium has tens of magnetosomes and its outer surface is covered in about 30 flagella (*Rodgers et al., 1991*). Cells within a consortium coordinate their activity to exert a net force that is parallel to the consortium's net magnetic moment (*Almeida et al., 2013*), which varies among individuals from 3.9 fAm$^2$ to 10.9 fAm$^2$, enabling them to swim at speeds ranging from 45 μm/s to 134 μm/s (*Petroff et al., 2022*).

MMB provide an apt system for the broader study of magnetotaxis and its evolution for three reasons. As these consortia are significantly larger than those studied in previous investigations (*Codutti et al., 2024*), it is comparatively simpler to track their motion through a pore space. Additionally, as these bacteria are taken directly from their natural habitat, their phenotypes do not reflect the process of domestication. These bacteria are readily enriched from tide pools in Little Sippewissett Salt Marsh (*Simmons and Edwards, 2007*; *Shapiro et al., 2011*; *Schaible et al., 2024*), near Falmouth, Massachusetts. Finally, the physical characteristics of their habitat can be directly measured. These bacteria live in the spaces between (*Martins et al., 2009*; *Simmons et al., 2007*; *Keim et al., 2007*), which we measure to have a mean diameter of 0.6 ± 0.16 mm. A random packing of such grains forms pores with a typical radius (*Andreotti et al., 2013*) of $r = 0.2 \pm 0.05$ mm. The local geomagnetic field (*NOAA, 2024*) is 50.9 μT, with an inclination of 65.6°.

## Results

*Figure 1a* shows the two-dimensional artificial pore space in which we track the motion of consortia. It is composed of a square lattice of convex pores. When a magnetotactic bacterium collides with a convex barrier, it must temporarily swim against the applied magnetic (which points from left to right in *Figure 1*) in order to find a passageway leading forward. Because the geometry of these convexities can be precisely controlled, the study of locomotion through this pore space provides a useful experimental system in which to study the dynamics of backtracking. While convex pores cannot be formed by convex particles (e.g. spheres), there is good reason to believe that backtracking is necessary for efficient navigation in natural sediment where clogging of pores creates pore-scale dead ends and convexities. Biofilms growing between arrays of circular posts in a microfluidic device are observed to connect neighboring posts (*Kurz et al., 2023*; *Hassanpourfard et al., 2016*; *Gaol et al., 2021*), thus forming barriers that include both convex and concave sections and create dead ends in the pore space. Similar features are found in three-dimensional simulations of biofilm growth between spherical particles (*Peszynska et al., 2016*; *von der Schulenburg et al., 2009*). Additionally, muddy pore water contains many small cohesive particles. When these particles move through narrow passageways, they clog (*Dressaire and Sauret, 2017*; *Yin et al., 2024*) and create boundaries similar to those shown in *Figure 1a*. Air bubbles can also create pore-scale dead ends that would require backtracking (*Wang et al., 2021*). The concentration of convex traps present in the pore network analyzed here is almost certainly much greater than in any natural habitat. Nonetheless, as magnetotactic bacteria constantly swim through the pore space, they will inevitably encounter convexities and pore-scale dead ends. Efficient navigation thus requires the ability to backtrack. In order to understand the phenotypic traits that allow efficient backtracking, we choose to analyze a pore space with many convex traps.

An applied magnetic field directs MMB through the pore network. The field is oriented 45° off the primary axis of the lattice (i.e. from left to right in *Figure 1*) to ensure that every field line intersects the walls of a pore. Note that a square lattice of circular pores would align the shortest path through the pore, from the entrances on the left to the exits on the right in *Figure 1*, with the magnetic field.

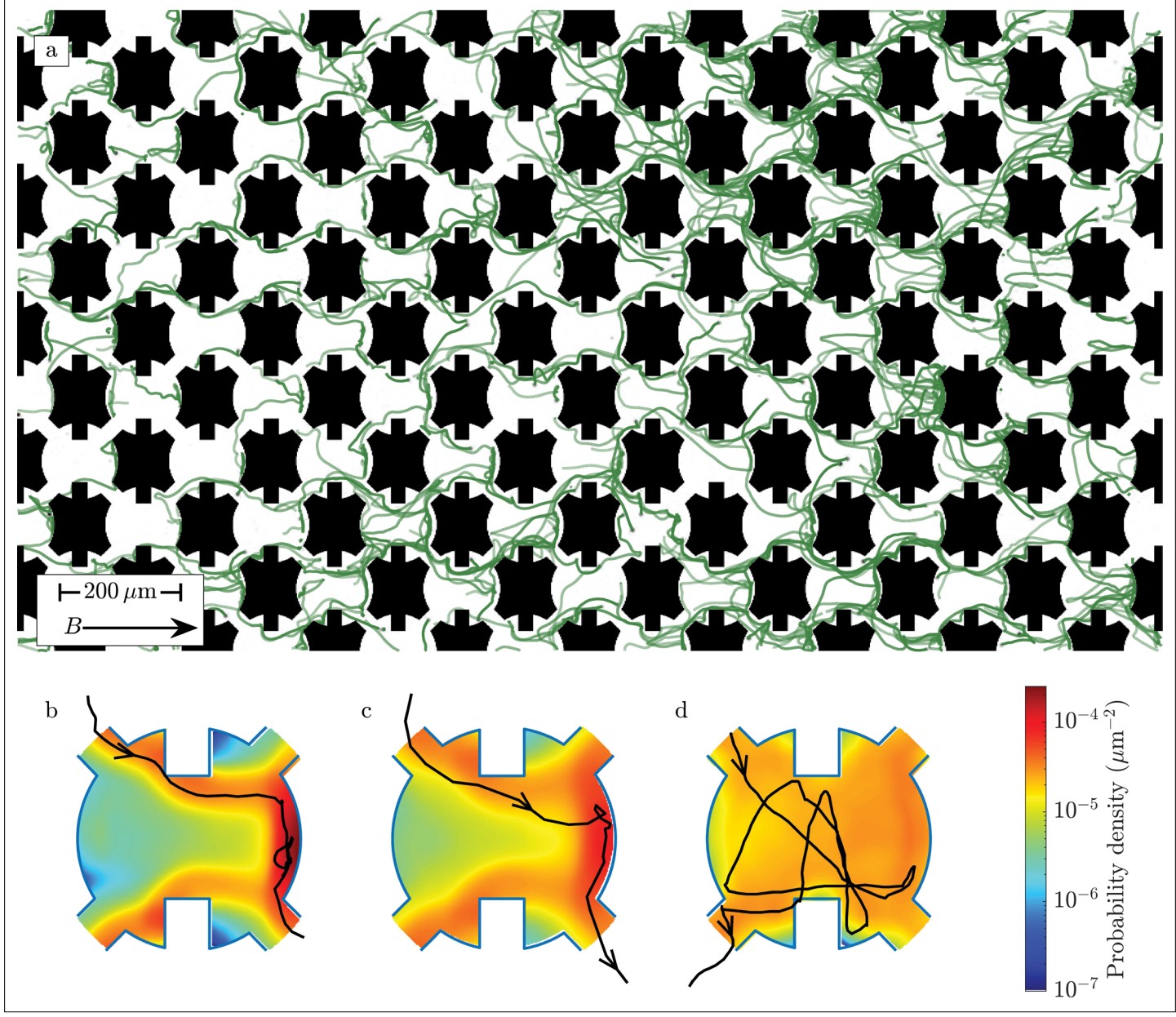

**Figure 1.** Multicellular magnetotactic bacterial consortia are directed through an artificial pore network by an applied magnetic field. (**a**) The trajectories (green lines) of several hundred consortia (black dots) are shown. While each consortium is composed of tens of individual bacteria, it grows and moves like a single organism. The black boundaries show the positions of the microfluidic pillars, which separate the pores. The applied magnetic field is $B = 940$ µT, corresponding to Sc = 0.31. (**b**) Averaging the positions of consortia across all pores over the course of an experiment yields the probability density within a pore. The edges of the pore are highlighted in blue. A detailed description of how these probability distributions are measured is provided in Materials and methods. The magnetic field is $B = 3500$ µT and the scattering number Sc = 0.08. The black line shows a representative trajectory of a consortium, which passed through a pore in 5.6 s. (**c**) Weakening the magnetic field produces a wider distribution of positions, which extends from the northernmost wall to the passages to neighboring pores. This panel corresponds to the experiment in (**a**). A representative trajectory is shown for a consortium that escaped in 1.2 s. (**d**) At low magnetic field ($B = 75$ µT, Sc = 3.85), consortia swim in roughly straight lines and are randomly reoriented by collisions with the walls. The black line shows the trajectory of a consortium that escaped to a southward pore after 7.4 s.

To remove this unrealistic artifact of a highly symmetric pore space, we add small rectangular barriers to the top and bottom of each circular pore.

We observe that as the applied magnetic field is reduced, the consortia explore an increasingly larger fraction of the pore. At very high magnetic fields (*Figure 1b*, *Video 1*), consortia cannot turn

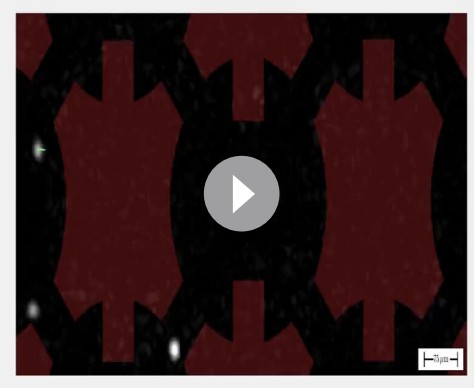

**Video 1.** Multicellular magnetotactic bacteria move through an artificial pore space. Each white dot shows a spherical consortium composed of tens of individual bacteria, which grow and move like a single multicellular organism. The red pillars show the separations between neighboring pores. The video is slowed to 80% of the true speed.

https://elifesciences.org/articles/104797/figures#video1

**Video 2.** At scattering number $\mathrm{Sc} = 3.85$, trajectories of consortia (bright lines) are roughly straight between collisions. The video is slowed to 80% of the true speed.

https://elifesciences.org/articles/104797/figures#video2

away from a field line and so cannot move around an obstruction from one pore to the next. As they become trapped against pore boundaries for prolonged periods, magnetotaxis is slowed. At an intermediate magnetic field (*Figure 1c*, *Video 1*), consortia concentrate near the connections that lead northward and rapidly move through the network. At low magnetic field (*Figure 1d*, *Video 1*), consortia are uniformly distributed through the pore and often escape southward. In the limit of vanishing magnetic field, the motion is unbiased and magnetotaxis is obviously impossible. Consequently, we expect that maximizing the average speed $U_{\mathrm{drift}}$ of a consortium through a given pore space requires a balance between magnetotaxis and obstacle avoidance. If consortia align with the magnetic field too quickly or too slowly, this balance is broken, and the organism becomes trapped or wanders randomly.

Dimensional analysis (*Bridgman, 1922*) gives insight into how this balance is reflected in the phenotype of an arbitrary magnetotactic bacterium and its environment. For pores of a given shape, the important aspects of the environment are described by the ambient magnetic field intensity $B$ and the pore radius $r$. Five aspects of the phenotype may effect $U_{\mathrm{drift}}$. They are as follows: the swimming speed $U_0$, the magnetic moment $m$, the length $a$ of the organism, its rotational diffusion coefficient $D_{\mathrm{rot}}$, and its rotational hydraulic mobility $\mu_{\mathrm{rot}}$. The mobility $\mu_{\mathrm{rot}} \propto 1/(\eta a^3)$—where $\eta$ is the viscosity of water and the proportionality constant is determined by the shape of the organism—relates to torque on the organism to its angular velocity. As these eight variables share four physical dimensions (charge, length, time, and mass), it follows that the efficiency $U_{\mathrm{drift}}/U_0$ can be expressed as a function of three dimensionless combinations of phenotypic and environmental variables.

We call the first, and most important, of these dimensionless numbers the scattering number

$$\mathrm{Sc} = \frac{U_0}{r\gamma}, \tag{1}$$

where $\gamma = \mu_{\mathrm{rot}} m B$ is the rate at which a magnetotactic bacterium aligns with the ambient magnetic field. Sc has two useful and equivalent interpretations. As bacteria are rapidly scattered by collisions with surfaces (*Drescher et al., 2011*; *Petroff and McDonough, 2024*; *Lauga, 2016*), it compares the rate $U_0/r$ at which an organism is randomly reoriented by collisions, to the rate at which it aligns with the magnetic field. Alternatively, a magnetotactic bacterium swims a distance of order $U_0/\gamma$ in the time required for it to realign with the magnetic field after a random reorientation. Consequently, if $\mathrm{Sc} < 1$, Sc is the typical fraction of the pore that is explored by a magnetotactic bacterium. As described in Materials and methods, we measure Sc for each enrichment of consortia immediately before it enters the pore space.

The next dimensionless number is $a/r$. One expects that cell lengths $a = 1 - 20\,\mu\mathrm{m}$ are small compared to the size of pores they swim through. While papers rarely report the grain size of the sediment from which magnetotactic bacteria are enriched, it is reasonable to assume that the tidal ponds, marshes, estuaries, and rivers are composed of grains similar to medium or coarse sand. Such is the case in the ponds of Massachusetts, where we enrich MMB and measure $r = 0.2 \pm 0.05\,\mathrm{mm}$. As $a/r \approx 0.05$, we conclude that the excluded volume of bacteria does not meaningfully affect their ability to navigate. In what follows, we assume that unreported pore radii are within a factor of 2 of $0.2 \pm 0.05\,\mathrm{mm}$.

The final dimensionless number is $D_{\mathrm{rot}}r/U_0$. This ratio compares the rate that a magnetotactic bacterium is reoriented by rotational diffusion to the rate it is reoriented by collisions. The importance of rotational diffusion on the motion of *Magnetococcus marinus* through a pore space has been previously examined (*Dehkharghani et al., 2023*). In natural sediment, $D_{\mathrm{rot}}r/U_0$ takes values of 0.4 ± 0.1 for MMB and, we estimate (*Dehkharghani et al., 2023*), 0.08–0.3 for *M. marinus*. In the experiments presented here, the pore sizes are smaller than in nature and $0.15 < D_{\mathrm{rot}}r/U_0 < 0.25$. *Figure 1d* and *Video 2* show roughly straight trajectories at low magnetic field. As $D_{\mathrm{rot}}r/U_0$ is reasonably small, we conclude that the largest changes in orientation are the result of collisions with walls rather than rotational diffusion.

Thus, we find that magnetotactic bacteria can be treated as point-like swimmers that turn deterministically to align with the ambient magnetic field and are randomly scattered by collisions with the pore boundaries. In this limit

$$\frac{U_{\mathrm{drift}}}{U_0} = f(\mathrm{Sc}),\tag{2}$$

where $f$ is an unknown function.

We proceed to measure $f(\mathrm{Sc})$ from the rates at which MMB transition between pores. To better visualize these pore-scale dynamics, we reduce the full pore network (*Figure 1a*) to the miniature network shown in *Figure 2a*. We define $k_+$ as the rate at which consortia move in the direction of the magnetic field from one pore to another. The corresponding rate for motion against the field is $k_-$. As shown in Appendix 2, the continuum limit of this pore network yields

$$f(\mathrm{Sc}) = \alpha\frac{2r}{U_0}(k_+ - k_-),\tag{3}$$

where the geometric prefactor $\alpha = 0.8344$ is determined by the ratio of the lattice spacing to pore size. We measure $k_+$ and $k_-$ from the number $N(t)$ of consortia in a given pore as a function of time $t$ and the times that consortia move to neighboring pores. If we observe $n_+$ transitions to a northward pore in a certain amount of time $T$, the best estimate of $k_+ = n_+(\int_0^T N(t)\mathrm{d}t)^{-1}$. We estimate $k_-$ in a similar manner. We experimentally vary the scattering number by adjusting the magnitude of the applied magnetic field.

*Figure 2b* shows how the transition rates vary as Sc increases from 0.14 ($B = 5000$ µT) to 9.3 ($B = 75$ µT). The rate $k_+$ of northward motion is non-monotonic and is maximized at a value of Sc where consortia concentrate near the connectors to the northward pores (see *Figure 1c*). The rate of southward migration increases monotonically with Sc as consortia explore an ever-growing fraction of the pore space. The drift velocity (*Figure 2c*) is non-monotonic and reaches its maximal value of $\approx 0.25 U_0$ at a critical scattering number of $\mathrm{Sc_c} \approx 0.4$.

One can crudely estimate the critical scattering number directly from the pore geometry. As the scattering number gives the typical fraction of the pore that a consortium explores, we expect that $\mathrm{Sc_c} \sim \delta/r$, where $\delta$ (see *Figure 2a*) is the distance between the northernmost pore wall and the exit to the next pore. In this pore network, $\delta/r = 1 - 2^{-1/2} \approx 0.29$, which is similar to $\mathrm{Sc_c}$.

Next, we compare these measurements to the motion of simulated consortia. In light of the dimensional analysis presented above, we approximate the consortia as points that swim parallel to their magnetic moments at a constant speed $U_0$. The angle $\theta$ between the direction of motion and the magnetic field evolves in time $t$ as $\mathrm{d}\theta/\mathrm{d}t = -\gamma\sin(\theta)$. We assume that, after a collision with a boundary, the angle at which consortia escape the wall is uniformly distributed. Non-dimensionalizing time by $1/\gamma$ and distances by $r$ reveals the dimensionless swimming speed to be Sc. We simulate $10^3$ consortia in the pore geometry shown in *Figure 2a* and calculate the time it takes each to escape to a neighboring

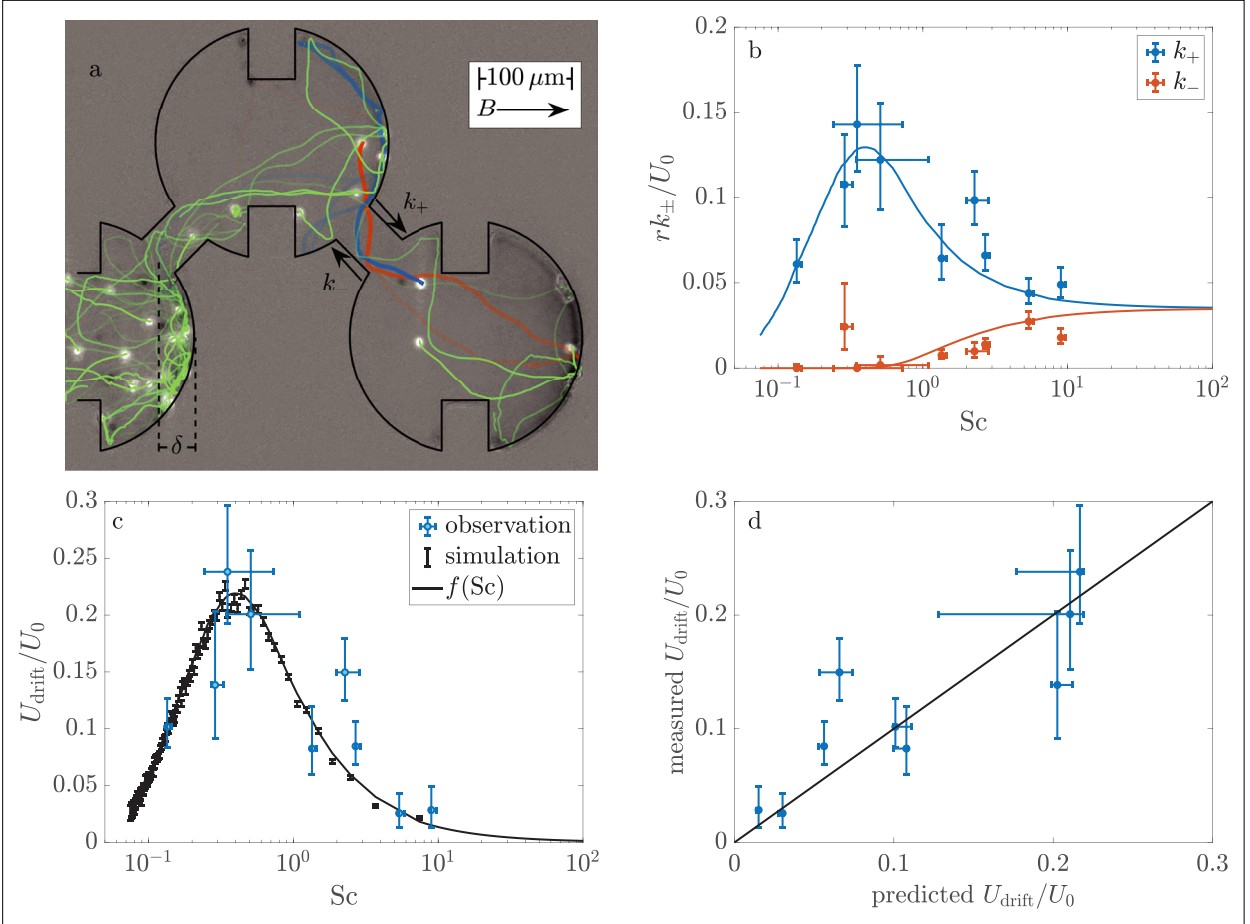

**Figure 2.** The speed of consortia through a pore space is maximized at a finite value of Sc. (**a**) The rates $k_\pm$ at which consortia (white spots) move with ($k_+$) and against ($k_-$) the applied magnetic field are measured in a small network of pores. The trajectories—shown here as colored lines that dim over the course of 4 s—are reconstructed from the instantaneous positions of consortia. Consortia move less than a consortium radius between frames, which are recorded at 75 frames/s. The blue and red trajectories highlight two consortia that transition either in the direction of $B$ (blue) or in the negative sense (red). This image, taken at $\text{Sc} = 1.98 \pm 0.36$ ($B = 207\ \mu\text{T}$), is a still from *Figure 2—video 1*. (**b**) Tracking the motion of a total of 938 consortia at various magnetic fields provides $k_\pm(\text{Sc})$. The reported values of Sc are measured for each group of consortia moments before it enters the pore space. The solid lines show the predicted relationship for simulated consortia. (**c**) The asymmetry in transition rates causes multicellular magnetotactic bacteria (MMB) to drift through the pore space in the direction of the magnetic field at speed $U_{\text{drift}}$. The solid line shows the predicted relationship. The theoretical curves in (**b**) and (**c**) require no fitting parameters. Their calculation from simulations is discussed in Materials and methods. The source data for the experiments and simulations can be found in *Figure 2—source data 1* and *Figure 2—source data 2*, respectively. (**d**) The measured values of the drift velocity are well approximated by the predicted form of $f(\text{Sc})$ with no fit parameters. The horizontal error bars reflect both the uncertainty in the measured value of Sc and the uncertainty in $f(\text{Sc})$ arising from the finite number of simulations, which are discussed in Materials and methods.

The online version of this article includes the following video and source data for figure 2:

**Source data 1.** Experimental data for transition rates of multicellular magnetotactic bacteria (MMB) between chambers.

**Source data 2.** Theoretical data for the transition rates between chambers.

**Figure 2—video 1.** Video of a small pore space in which $k_\pm$ are measured.

https://elifesciences.org/articles/104797/figures#fig2video1

**Figure 2—video 2.** A simulation of point-like consortia (red dots) moving between pores shows the equilibrium distribution and fluctuations at $\text{Sc} = 1.25$.

https://elifesciences.org/articles/104797/figures#fig2video2

pore. As discussed in Materials and methods, these first passage times are exponentially distributed and the decay constants give $k_+$ and $k_-$ for any particular choice of Sc. *Figure 2—video 2* shows a representative simulation. Repeating this process for 100 choices of Sc provides predictions for $k_+$, $k_-$, and $f(\text{Sc})$ (see *Figure 2c*) with no free parameters. In the particular geometry shown in *Figure 2a*, we predict that the drift velocity is maximized at a critical scattering number of $\text{Sc}_c = 0.40$, which is found

by fitting the maximum of the smooth curve in *Figure 2c* to a parabola. Details of these simulations are provided in Materials and methods.

The good agreement (*Figure 2d*) between the predicted transition rates and our measurements leads us to accept the hypothesis that the magnetotactic efficiency $U_{\mathrm{drift}}/U_0$ is maximized at a finite value of the scattering number. We next ask if natural consortia are optimized for their native environments.

Answering this question requires an estimate of the critical scattering number for natural pore geometries. The shape of the smooth function $f(\mathrm{Sc})$ shown in *Figure 2c* is generic but not universal. It necessarily goes to zero at very high scattering number (e.g. $B = 0$, see *Figure 1d*), where magnetotactic organisms do not align with the magnetic field. It similarly vanishes if $\mathrm{Sc} = 0$ (e.g. $B = \infty$, see *Figure 1b*), where organisms cannot swim around obstructions. These limits imply a global maximum at a critical value of $\mathrm{Sc_c}$, where the random fluctuations in swimming orientation due to collisions with boundaries regularly direct consortia toward northward pores, but not so great that consortia routinely escape to southward pores.

We estimate that, in natural sediment, $0.1 \lesssim \mathrm{Sc_c} \lesssim 1$. As Sc gives the typical fraction of the pore that a consortium explores and passageways leading north are most likely concentrated along the north-facing pore wall, we expect that the critical scattering number should be slightly less than 1. Moreover, a consortium cannot explore a pore if the distance $U_0/\gamma$ it is scattered after collisions is smaller than its body size $a$ (*Petroff et al., 2022*). It follows that $\mathrm{Sc_c} > a/r \sim 0.05$. Appendix 3 derives a similar estimate of $\mathrm{Sc_c}$ proceeding from the assumption that the locations of connections between pores are randomly distributed.

In their natural habitat, consortia swim through pores with a typical radius $r \approx 0.2\,\mathrm{mm}$ and align with a local geomagnetic field of magnitude $B_{\mathrm{geo}} = 50.9\,\mu\mathrm{T}$. As described in Materials and methods, we measure the average geomagnetic turning rate $\langle \gamma_{\mathrm{geo}} \rangle = 1.1 \pm 0.2\,\mathrm{s}$ and average swimming speed $\langle U_0 \rangle = 132 \pm 7\,\mu\mathrm{m/s}$ of 31 individual consortia. These values correspond to a population-averaged scattering number of 0.6±0.2, which is consistent with optimal navigation. There is substantial variability across individuals. While the majority of individuals fall in the predicted range and the smallest scattering number is 0.2, five individuals display scattering numbers between 2 and 3.2. Our theory cannot explain the tail of this distribution. It is plausible that a minority of consortia are close to division (*Keim et al., 2004*) and are unusually large and thus slowly turning. Alternatively, this variability could be a reflection of the polydisperse grains in which these consortia live.

Several years ago, we described (*Petroff et al., 2022*) the motion of consortia that were enriched from the same pond as those described here. These consortia were systematically slower, with a typical swimming speed of just 75 μm/s. Nonetheless, the measured values for $U_0$, $m$, and $a$ of these consortia correspond to $\mathrm{Sc} = 0.4 \pm 0.3$. We conclude that, despite the substantial phenotypic variability of this population across time, the scattering number is nearly constant and remains consistent with optimal navigation.

Finally, we turn our attention to the phylogenetic diversity of magnetotactic bacteria. A literature review (*Petersen et al., 1989*; *Spring et al., 1993*; *Pan et al., 2009*; *Nadkarni et al., 2013*; *Bahaj et al., 1996*; *Esquivel and De Barros, 1986*; *Acosta-Avalos et al., 2019*; *Petroff et al., 2022*; *Carvalho et al., 2021*) reveals tremendous phenotypic variability across taxa and environment. As described in Appendix 1, species from three phyla are included from latitudes between 22° S and 54° N. The geomagnetic field strengths across these latitudes range from $B_{\mathrm{geo}} = 23\,\mu\mathrm{T}$ with an inclination of -42°, in Rio de Janeiro, Brazil, to $B_{\mathrm{geo}} = 50.4\,\mu\mathrm{T}$ with an inclination of 69° near Grimmen, Germany. The sizes (1–18 μm) and speeds (12–141 μm/s) of magnetotactic bacteria both vary by an order of magnitude. The magnetic moments (0.3–54 fAm²) of the species vary by two orders of magnitude.

Despite this enormous variability across species, nearly all of these magnetotactic bacteria display scattering numbers that are within the expected range of $0.1 \lesssim \mathrm{Sc_c} \lesssim 1$. *Figure 3a* shows the scattering numbers for these species. These estimates use the local geomagnetic field and a pore size of medium to coarse sand. The average scattering number is 0.58, and individual values vary from 0.1 to 1.9.

As the scattering number is determined by the product of three phenotypic variables that all vary by more than an order of magnitude, it is surprising that the scattering numbers are as narrowly distributed as they are. To quantify this peculiarity, we consider an unbiased null model of phenotypic diversity. We assume that collision rates $U_0/r$, hydrodynamic mobilities $\mu_{\mathrm{rot}}$, and geomagnetic torques

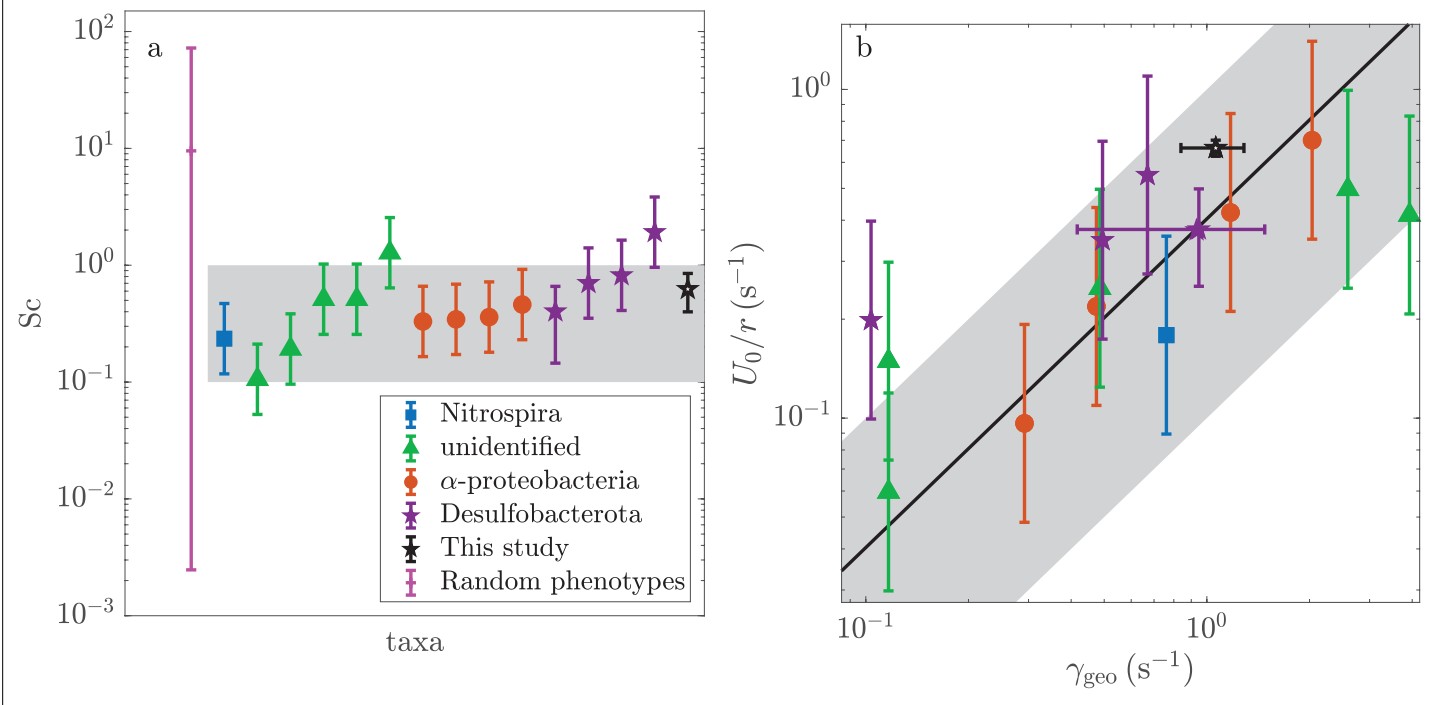

**Figure 3.** Diverse magnetotactic taxa are able to optimally navigate the pore spaces of medium to coarse sand. These organisms tune their swimming speeds, rotational hydrodynamic mobilities, and magnetic moments to the local geomagnetic field. A table of the phenotypic parameters shown here and their sources is provided in Appendix 1. (**a**) All taxa have scattering numbers that are similar to or slightly less than unity. By contrast, random phenotypes, which lack correlations between phenotypic variables, are characterized by large and widely distributed scattering numbers. That the measured values of Sc are anomalously narrowly distributed indicates selective pressure. The gray shaded region ($0.1 < \mathrm{Sc} < 1$) shows the estimated range of scattering numbers that allow for efficient navigation. (**b**) The scaling analysis predicts that the rate that a species aligns with local geomagnetic field is proportional to the swimming speed. The solid line shows $U_0/r\gamma_{\mathrm{geo}} = 0.40$, which is the optimal value predicted in *Figure 2c*. The gray shaded region and legend are the same as in the first panel.

$mB_{\mathrm{geo}}$ are independently tuned to the local environment. The distributions of these traits reflect the range of biologically accessible phenotypes and the relative abundances of ecological conditions that select for the particular value of each trait. We randomly generate ecologically plausible phenotypes through a bootstrapping procedure; we sample from the measured phenotypic variables with replacement.

When compared to random phenotypes, the scattering numbers of real magnetotactic bacteria are anomalously small and narrowly distributed. The average scattering number of randomly generated phenotypes is 9.5 with 95% of these values falling between $2 \times 10^{-3}$ and 72.8. Thus, in the absence of selective pressure, we would expect to find substantially more variability in the scattering numbers between species. Most of these species would move diffusively through the pore space. The probability that 15 random phenotypes fall within the measured range of naturally occurring scattering numbers is $2.5 \times 10^{-6}$.

We conclude that natural selection has tuned the scattering numbers of magnetotactic bacteria in a way that is consistent with optimal navigation. This tuning is reflected in correlations between the speeds, sizes, shapes, and magnetic moments of magnetotactic bacteria, which are absent in the random phenotypes.

The scaling analysis also predicts the specific correlation between phenotypic variables. Optimal navigation requires the scattering rate $U_0/r$ of each species to be proportional to its geomagnetic alignment rate $\gamma_{\mathrm{geo}}$. As shown in *Figure 3b*, these rates are indeed proportional. The correlation coefficient is 0.77. The probability that the scattering numbers of 15 random phenotypes show correlations that are this large is $4 \times 10^{-4}$. Note that if flagella and magnetosomes compete for resources or energy, then $U_0$ and $\gamma_{\mathrm{geo}}$ would be negatively correlated. As discussed in Appendix 4, correlations between these phenotypic parameters do not reflect the dependence of each upon body size.

There is clearly substantial scatter in *Figure 3*. This scatter likely reflects two aspects of magneto-tactic motility that do not generalize well from the study of MMB. First, the orientation of swimming MMB is decorrelated primarily by collisions with boundaries, and the effect of rotational diffusion can be ignored. It is likely that rotational diffusion may be more important for smaller magnetotactic bacteria. In such cases, the critical scattering number becomes a function of $D_{rot}r/U_0$. Additionally, when an MMB collides with a surface, the angle at which it escapes can be approximated as uniformly distributed. If microbes swim along surfaces (*Ostapenko et al., 2018*; *Codutti et al., 2022*) for a distance $\ell$ and escape at typical angles (*Kühn et al., 2017*; *Petroff and McDonough, 2024*), then the critical scattering number also depends on $\ell/r$ and the moments of the escape angle distribution. There is evidence of these effects in *Figure 3*. The Desulfobacterota live at scattering numbers of 0.9±0.3 that are somewhat greater than those of the α-Proteobacteria, which live at scattering number 0.37 ±0.03. It is plausible that this difference reflects taxonomic differences in rotational diffusion and cell-wall interactions. However, as the scattering numbers of these bacteria remain within the expected range of values, it seems that these corrections are of secondary importance.

We conclude that all magnetotactic bacteria discussed here are, to a good approximation, equally and optimally efficient navigators of their native pore spaces.

## Discussion

We have found that in an ecologically relevant limit, the efficiency of magnetotaxis is a function of a single dimensionless combination of environmental and phenotypic parameters. The optimal pheno-type swims a distance that is similar to the pore radius before aligning with the magnetic field. Doing so causes magnetotactic bacteria to concentrate in the fraction of the pore where escapes are most likely to be found. Because the efficiency of magnetotaxis depends only on the ratio of phenotypic and environmental parameters, it can be carried out equally well by magnetotactic bacteria with different phenotypes. This degeneracy is found to accommodate the natural diversity of magneto-tactic bacteria.

As the magnetic moment $m = U_0/(\text{Sc}_c r \mu_{rot} B_{geo})$, the results presented in *Figure 3* are equivalent to showing that this analysis predicts the magnetic moments of unrelated magnetotactic bacteria across two orders of magnitude. This rescaling is difficult in practice as the rotational hydraulic mobility $\mu_{rot}$ of the organisms can only be roughly estimated from their shapes. The magnitude of the magnetic moment is not determined by the rotational diffusion of the magnetotactic bacteria, which we find to be of secondary importance. Rather, as has been previously suggested (*Dehkharghani et al., 2023*), the orientation of a swimmer is decorrelated primarily by collisions with the pore boundaries. Our results agree well with previous work (*Codutti et al., 2024*; *Petroff et al., 2022*), which shows that too great a magnetic moment (Sc ≪ Sc$_c$, see *Figure 1b*) is as detrimental to magnetotaxis as a magnetic moment that is too small (Sc ≫ Sc$_c$, see *Figure 1d*).

Insofar as the ability to quickly move through the pore space aids bacteria in reproducing (*Smith et al., 2006*), $f(\text{Sc})$ can be considered a one-dimensional slice of a fitness landscape (*Bank, 2022*). *Figure 2* shows that this slice is smooth and displays a single global maximum. Reaching this maximum requires natural selection to tune a species' phenotype to reflect its environment such that $U_0/\mu_{rot}m = \text{Sc}_c B_{geo}r$. This result suggests that if a bacterium acquires the genes for magnetosome formation and alignment—either de novo or through horizontal gene transfer—then its magneto-tactic efficiency is immediately nonzero. The smoothness of $f(\text{Sc})$ suggests that natural selection can quickly guide the phenotypic variability in size, swimming speed, and magnetic moment to greater efficiencies. Studies in which the genes for magnetosome formation are added to non-magnetotactic bacteria (*Kolinko et al., 2014*; *Dziuba et al., 2024*) could provide a strong test of this prediction.

Efficient magnetotaxis imposes a trade-off between size (through $\mu_{rot}$), speed, and magnetic moment. A magnetotactic bacterium can be fast swimming and large only if it produces a very large number of magnetosomes. The constraint that $U_0/r\gamma_{geo} = \text{Sc}_c$ defines a two-dimensional surface in a three-dimensional space of phenotypes. Any species with phenotypic variables laying on this surface is optimally efficient. This degeneracy allows a species to remain optimally efficient at magnetotaxis while tuning any two of its three phenotypic parameters to other constraints imposed by its ecology. The phenotypic flexibility provided by this degeneracy gives insight into how magnetotaxis became so widely distributed across bacteria with diverse phylogenies, morphologies, and ecological roles.

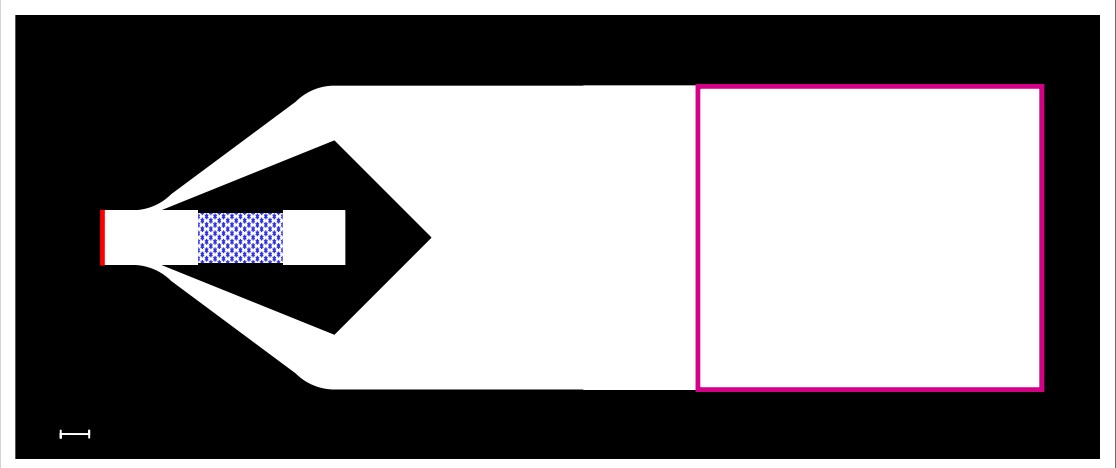

**Figure 4.** Schematic of a typical microfluidic photomask. White regions correspond to 40 µm tall regions that are filled with seawater. The pink area shows the location where we place a spacer that creates a 1 ml void, called the vestibule. After filling the chamber with filtered seawater, an enrichment is inoculated into the vestibule. A 300 µT field pointed to the left directs consortia to accumulate near the red line, where Sc is measured. Reversing the magnetic field directs consortia into the pore network, which is highlighted here in blue. The scale bar is 1 mm.

## Materials and methods

### Enrichment of MMB

We enrich MMB from the environment using established methods (*Shapiro et al., 2011*; *Petroff et al., 2022*; *Schaible et al., 2024*). We collect sediment from a shallow pool in a Massachusetts salt marsh ($41°34'34.2''$ N, $70°38'21.4''$ W) between late spring and early fall and maintain this sediment in the lab. To concentrate MMB, we gently agitate the sediment and position a neodymium magnet near the surface. After allowing 20 min for cells to respond, we extract 1 ml of water from the region adjacent to the magnet. A secondary enrichment, described below in Step 3 of 'Loading the microfluidic chamber', is performed within the microfluidic device. Once brought to the lab, the sediment continues to produce substantial numbers of MMB for several months.

### Geometry of the pore space

The pore spaces shown in *Figures 1 and 2* are produced using standard photolithography techniques (*Whitesides et al., 2001*). We use the negative epoxy photoresist SU-8 2050 (Kayaku Advanced Materials) to form a mold for the chambers. PDMS (Dow Sylgard 184) is poured over the photomask to form the microfluidic chip, which is attached to a microscope slide. *Figure 4* shows a typical photomask used in our experiments.

All pore spaces have a height of 40 µm. We found that this height keeps all consortia in the focal plane of the Zeiss 5x objective. Because the height of the chamber is significantly greater than the diameter of a consortium, the consortia remain fairly dilute even at high magnetic field. All pore networks are laid out on a square lattice of pores. We experimented with several pore geometries. We attempted to vary the shape of a pore from circular (as shown in *Figures 1 and 2*) to make the north and south facing pore boundaries slightly more or less curved. These changes were found to be too subtle as their effect on $k_{\pm}$ is small relative to the measurement error in the rates. The rates measured in these slightly different geometries are averaged together. We varied the pore radius from 25 µm to 150 µm. We found that chamber radii between 100 µm and 125 µm provided the greatest experimental control. Smaller chambers quickly become densely packed with consortia, especially in the summer months when consortia are most abundant. Larger chambers require very weak magnetic fields to probe high scattering numbers ($B = U_0/r\mu_{\rm rot}m{\rm Sc}$). As we seek to precisely tune the scattering number, the maximum chamber radius is limited by the spatial inhomogenities (~5 µT) in the magnetic field produced by the objective.

## Loading the microfluidic chamber

MMB are loaded into the microfluidic pore space in five steps. These steps ensure that: (1) the pores are initially free of all microbes and chemical gradients, (2) the consortia entering the pores are at very high purity, and (3) the scattering number of a consortia is directly measured, moments before the start of the experiment. The microfluidic device shown in *Figure 4* makes it possible to meet all three of these criteria.

We reduce the time between these steps to about 7 min to minimize the number of consortia that arrive to the pore space. Doing so provides a dilute concentration within the pore space and limits interactions between consortia. However, this procedure selects strongly for the fastest swimming consortia, which are not representative of the natural community.

### Step 1: Building the vestibule

Before pouring the PDMS over the photoresist, we place a 7 mm tall right triangular prism, which we 3D print, on the wafer (see *Figure 4*). After baking the PDMS, this prism is removed to form a 1 ml void space, which slopes toward the chambers. Following Step 2, we inject the enrichment of consortia into this void space, which we call the 'vestibule'. Its large size and distance from the chambers reduce the number of non-magnetotactic microbes that make their way into the pore network.

### Step 2: Filling the pore space

To prevent the introduction of microbes into the pore space before the start of the experiment, the microfluidic device is first filled with filtered (0.2 μm) seawater, which is collected from the salt marsh where the consortia grow.

### Step 3: Secondary enrichment

Consortia are first enriched using the same method as described by *Schaible et al., 2024*. This procedure produces a reasonably high purity enrichment of MMB; however, a small number of ciliates and other sediment bacteria often remain in the fluid, which complicates image processing and tracking. This enrichment is loaded into the vestibule of the microfluidic chamber. A 300 μT field directs consortia from the vestibule to accumulate near the wall facing the pore space. The consortia begin accumulating a minute or two after the magnetic field is applied.

### Step 4: Measurement of Sc

The concentration of consortia decays exponentially from the wall with a decay length of $\lambda = 1.5 U_0/\gamma$ (*Petroff et al., 2022*). We measure $\lambda$ and, knowing $r$ for the particular choice of pore space, calculate the scattering number $\mathrm{Sc}_{\mathrm{load}} = \lambda/1.5r$ at $B_{\mathrm{load}} = 300$ μT. This measured scattering number is rescaled to the particular experiment at field $B$ as $\mathrm{Sc}_{\mathrm{load}} B_{\mathrm{load}}/B$.

To measure $\lambda$, we take three images in quick succession (75 frames/s). We track the motion of consortia between these frames to measure their swimming speed and to distinguish consortia from the background. Repeating this process once every 5 s for 3 min provides several thousand instantaneous positions. We fit the histogram of positions relative to the wall to an exponential to find $\lambda$, which generally varies only slightly between experiments.

### Step 5: Start of the experiment

Immediately following the collection of images needed to measure $\lambda$, the microscope is focused on the pore space, and several images of the empty pore space are recorded. The magnetic field is adjusted to select a particular value of Sc. Reversing the direction of the magnetic field directs consortia into the pore network, where they are tracked. Additionally, the field reversal directs consortia that are not in the pore space back toward the vestibule. This design prevents consortia from arriving in the pore space partway through an experiment.

## Control of the magnetic field

A uniform magnetic field is produced by a three-axis system of Helmholtz coils which were custom-designed with Woodruff Engineering to fit on a Zeiss Axio Observer inverted microscope. Two sets of Helmholtz coils produce uniform magnetic fields that are parallel to the table top. The experiment sits

in the center of a solenoid, which produces a magnetic field that is normal to the table top. The coils are powered by three Kepco bipolar operational amplifiers, which supply constant current.

The magnetic field $\mathbf{B}$ at the center of the experiment is an affine transformation of the current supplied to each coil. We define $\mathbf{B} = \mathbf{MI} + \mathbf{b}$, where $I_j$ is the current passing through coil $j$, $\mathbf{M}$ is a 3×3 matrix with constant coefficients that are determined by the geometry of the coils and microscope, and $b_i$ is the $i$th component of the ambient magnetic field. To measure $\mathbf{M}$ and $\mathbf{b}$, we focus the microscope on a small ($0.82\,\text{mm} \times 0.82\,\text{mm}$) triple-axis magnetometer (Adafruit MMC5603) that is centered in the field of view of the microscope. We apply a current $I_i(t) = A_i \sin(2\pi f t)$, where $A_i$ is chosen to be as large as possible without saturating the magnetometer and $f = 1\,\text{s}^{-1}$ through each of the coils (one at a time) and record $\mathbf{B}(t)$. This procedure provides several thousand measurements of the magnetic field for various currents. We measure $\mathbf{M}$ and $\mathbf{b}$ as

$$\begin{pmatrix} \mathbf{M}^\top \\ \mathbf{b}^\top \end{pmatrix} = \left( \mathbf{I}(t)^\top \quad \mathbf{1} \right)^+ \mathbf{B}^\top(t), \tag{4}$$

where + represents the pseudoinverse and 1 is a row vector of ones. Finally, we multiply $\mathbf{M}$ and $\mathbf{b}$ by a rotation matrix $\mathbf{R}$, which rotates the measured magnetic field about three orthogonal axes by small angles. We choose these angles to minimize the off-diagonal components of the matrix $\mathbf{RM}$. This rotation corrects for any misalignment between the magnetometer and the system of coils. For example, the magnetometer chip is rotated by about 1° relative to the edges of the board it is mounted on. This calibration procedure allows one to cancel out the magnetic field due to the Earth and the microscope and produces a uniform magnetic field with a precision of a few μT in any direction to within less than 1°.

We found that the coils must be recalibrated after several experimental runs as the magnetic field produced by the microscope objective drifts. At the start of each experiment, we measure the angle between the walls of the microfluidic chamber and the axes of the Helmholtz coils. We rotate the applied magnetic field such that the magnetic field is oriented 45° relative to the principal axis of the pore space.

## Tracking of MMB

We track the MMB consortia moving through the pores using TrackMate (*Tinevez et al., 2017*) in ImageJ. Before directing the consortia into the pore space, we average five images of the pore space, which we take as the background. During the experiment, we record the positions of consortia at either 20 frames/s (*Figure 1*) or 75 frames/s (*Figure 2*). We subtract the background image from each of the frames and convolve the image with the Laplacian of a Gaussian filter, the width of which is chosen to match the typical diameter of a consortium. Finally, we specify regions of interest in the connections between the pores and, using TrackMate, reconstruct the trajectories.

We identify a transition from one pore to the next when a trajectory crosses a line that equally divides the connection between neighboring pores. We linearly interpolate the positions of consortia along their trajectories to resolve the crossing times with sub-frame rate temporal resolution. This tracking provides the times that consortia leave and enter each pore.

This tracking method is reasonably robust. However, when two consortia swim close together, their IDs may switch. We estimate from the movies that this mislabeling happens a few times in each experiment. Importantly, as these mislabeling errors do not affect the number of trajectories crossing the midpoint line of the connectors, these errors do not affect the measurements of $k_\pm$.

## Measurement of the distribution of consortia within pores

*Figure 1b and c* shows the probability that a consortium is found at any point in a pore. These distributions are measured from the images shown in *Video 1*. Consortia appear white against the dark background. We convert these data into the probability distributions that are shown in *Figure 1* in three steps.

We first find the locations of pixels in each frame that exceed a certain intensity threshold, which was chosen such that the apparent size of the consortia before and after thresholding is similar. Next, we make a change of coordinate systems from the lab frame to that of the pore. The instantaneous position of each consortium is measured relative to the center of the pore that it is in. Finally, the

pore is divided into 0.9 μm square pixels, and the fraction of time that the pixel intensity exceeds the chosen threshold is measured.

The distributions shown in *Figure 1* give the probability that a random spot in a pore is covered by a swimming consortium, rather than the probability that the center of a consortium passes over a particular point.

In a typical experiment, of the order of a thousand consortia are tracked for several thousand frames. The probability distributions are found from between 1.2 and 7.0 million measurements of the instantaneous positions of consortia.

## Measurement of transition rates

The rates $k_\pm(\text{Sc})$ are measured in the miniature pore network shown in *Figure 2*, rather than in the full network of *Figure 1*. The choice to move to a small network to measure the pore-scale dynamics solves several experimental difficulties.

The first problem is that, at relatively high magnetic fields, consortia accumulate so tightly together that it is difficult to distinguish individuals. As we measure $k_\pm$ from the number $N(t)$ of consortia within a pore and number of times that consortia transition between pores, it is vital that $N(t)$ be accurately known. Rather than tracking the consortia within a pore, we only track them in the connections between pores. Starting from an empty pore space, we measure $N(t)$ by integrating the fluxes into and out of it. Recording data only within this reduced pore network allows us to increase the frame rate from 20 s$^{-1}$ to 75 s$^{-1}$. We double the dimensions of the pores in order to extend the time that consortia remain in the passageways and also to ensure that consortia are dilute. With these modifications, consortia remain in the passages between pores for tens of frames and move less than one third of a consortium diameter between frames. This reasonably high temporal resolution allows us to stitch instantaneous positions into trajectories with little ambiguity.

The second experimental challenge results from the phenotypic variability among the consortia. It appears that consortia move through the full pore space (*Figure 1*) with slightly different drift velocities. This effect is apparent in *Video 1*, which shows a front of consortia moving through the pore space at the highest magnetic field. Consortia are trapped for longer durations at the tail than at the leading edge. We expect that the phenotypic variability within the population leads to a diversity of scattering numbers and thus a diversity of drift velocities. Consequently, in the full network, the rates $k_\pm$ measured in a given pore vary in time as different subpopulations enter it. The rates at which diverse consortia move between two pores better reflect the population-averaged scattering number, which we directly measure.

## Simulations of MMB motion

MMB consortia are approximated as points that swim at a constant speed $U_0$ in the direction of their magnetic moment. As the microfluidic pores are quasi two-dimensional, the trajectories of consortia are assumed to be two-dimensional. The trajectory of such a swimmer has an analytic solution

$$\theta(\tau) = 2\cot^{-1}\left(e^\tau \cot\left(\frac{\theta_0}{2}\right)\right),$$

(5)

where $\tau = \gamma t$ is dimensionless time and $\theta_0$ is the initial angle between the magnetic moment and the magnetic field. We nondimensionalize distances by the pore radius $r$. The dimensionless position $x(\tau)$ along the direction of the magnetic field is

$$x(\tau) = x(0) + \text{Sc}\log\left(\frac{1 + e^{2\tau}\cot^2\left(\frac{\theta_0}{2}\right)}{1 + \cot^2\left(\frac{\theta_0}{2}\right)}\right) - \text{Sc}\tau.$$

(6)

The dimensionless position $y(\tau)$ in the orthogonal direction is

$$y(\tau) = y(0) + 2\text{Sc}\tan^{-1}\left(e^\tau \cot\left(\frac{\theta_0}{2}\right)\right) - 2\text{Sc}\tan^{-1}\left(\cot\left(\frac{\theta_0}{2}\right)\right).$$

(7)

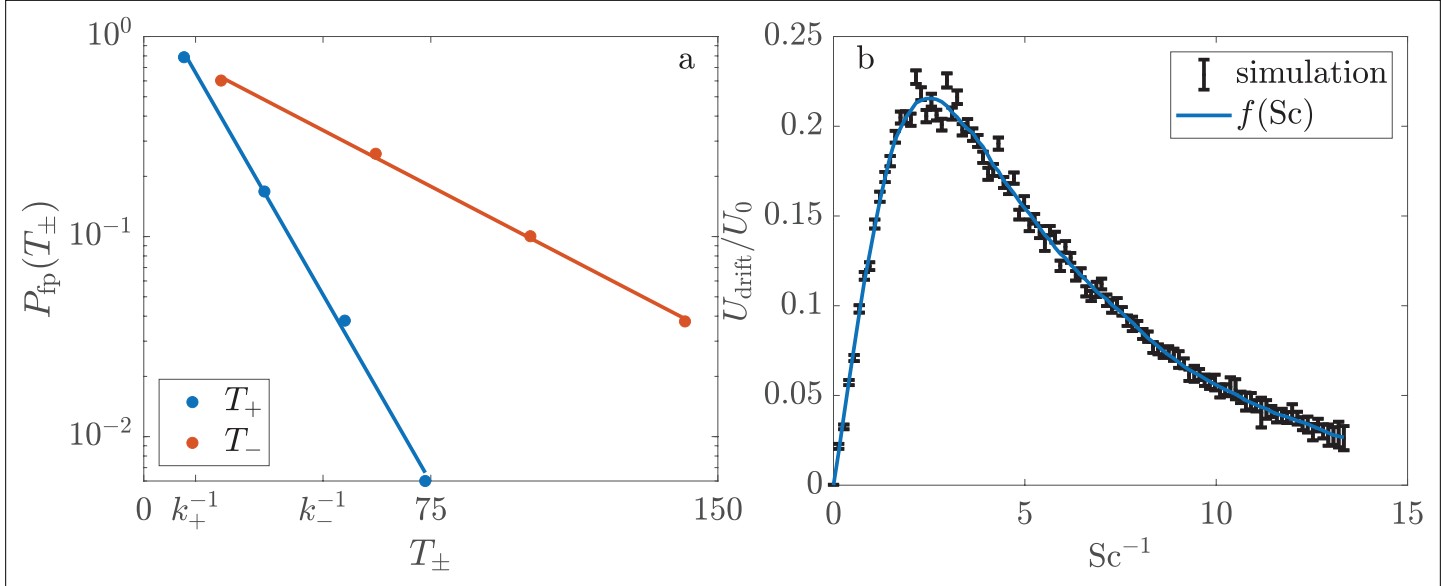

**Figure 5.** The transition rates $k_\pm$ and the drift velocity $U_{\text{drift}}$ are calculated from first passage times of simulated magnetotactic swimmers at various values of Sc. (a) The probability $P_{\text{fp}}(T_+)$ that swimmers escape a pore in the positive sense in a time $T_+$ is exponentially distributed. The blue dots are the results of simulations. The blue line shows the best fit exponential. The red dots and line correspond to motion against the direction of the magnetic field. These data were simulated at $\text{Sc} = 1.2375$. (b) The asymmetry in the transition rates causes swimmers to move through the pore space with an average speed $U_{\text{drift}}$, which is calculated for each simulation (black lines). The blue curve shows a most probable smooth function that approximates the results of the simulation. Only the smooth curves for $k_\pm$ are shown in *Figure 2*.

These simulations do not include the so-called 'escape motility' trajectories that have been observed in MMB (*Greenberg et al., 2005*; *Sepulchro et al., 2020*; *Yang et al., 2024*). Instances of this motion, where MMB briefly swim against the applied magnetic field, are visible in *Videos 1 and 2*; however, they do not lead consortia from one pore to another.

This analytic solution allows one to simulate the motion of a swimmer in a confined geometry without discrete time steps. For a given initial position, we solve *Equations 6 and 7* for the time $\tau$ at which the swimmer comes in contact with a boundary. We move the swimmer to the point of contact and randomly select a new value of $\theta_0$. We assume the distribution of escape angles is uniform over the range of values that prevent swimmers from moving through walls. We repeat these two steps for as long as is necessary. We record the times and locations of contacts between the swimmer and the boundaries.

We calculate $k_\pm(\text{Sc})$ in two steps. First, we simulate the motion of a swimmer that is confined to a single pore, which is identical in shape to those shown in the experiment and scaled to have radius 1. We run this simulation until the probability that the swimmer strikes any given section of the boundary ceases to evolve in time. This steady-state distribution $P(x, y)$ gives the probability that a thermalized swimmer collides with any given section of the pore boundary. Next, we randomly choose an initial position $(x_0, y_0)$ on the pore boundary from $P(x, y)$. We start a swimmer from this position with a random initial orientation. We solve for the collisions between the swimmer and pore boundary until the swimmer collides with a section of the boundary that would lead it to a neighboring pore. We record the times $T_+$ and $T_-$ that swimmers escape to northward and southward pores, respectively. Repeating this process a thousand times provides the distribution of first passage times, which is found to be exponential. *Figure 5a* shows the distribution of first passage times in a representative experiment. We report transition rates (*Figure 2*) that are nondimensionalized by the time $r/U_0$ for a swimmer to cross a pore radius. We measure these values in the simulations as $rk_\pm/U_0 = \langle T_\pm(\text{Sc})\rangle/\text{Sc}$. At very low scattering numbers, swimmers do not escape to southward pores in computationally accessible timescales. We extrapolate these rates from our simulations assuming that $rk_-/U_0 \propto \text{Sc}$ for $\text{Sc} \ll 1$. We exactly solve for the scattering rates in the limit of infinite Sc from similar code in which swimmers move in straight trajectories between collisions.

We repeat this process of choosing a specific value of Sc and calculating $k_+$, $k_-$, and $U_{\text{drift}}$ for 100 different values, which are shown in *Figure 5b*. Because we are able to simulate these dynamics at Sc $= \infty$, it is convenient to choose values of Sc$^{-1}$ that are uniformly spaced from 0 to a large value. We find that $U_{\text{drift}}/U_0 \approx 0.16\text{Sc}^{-1}$ if Sc$^{-1} \ll 1$. In the limit of Sc$^{-1} \gg 1$, $U_{\text{drift}}/U_0 \approx 0.58e^{-0.22/\text{Sc}}$.

Assuming that an analytical solution for the transition rates is smooth, large differences between rates that are simulated at similar values of Sc result from the finite number of swimmers simulated. We seek the smooth function $f(\text{Sc})$ that best approximates the simulations. Because we can exactly simulate the dynamics at Sc $= \infty$, it is convenient to estimate $f(\text{Sc}^{-1})$ rather than $f(\text{Sc})$. To find the most probable value of this function at a point $\text{Sc}_0$, we assume that the true function $f(\text{Sc}^{-1})$ can be approximated as parabolic over the interval $\text{Sc}_0^{-1} \pm \Delta$. It is straightforward to calculate the probability that a given test function would produce a mismatch between its predictions and the 1000 first passage processes that were simulated at each of scattering numbers in the range $\text{Sc}_0^{-1} \pm \Delta$. We find the most probable value of $f(\text{Sc})$ and the 95% confidence interval at 200 test points. The width of the confidence interval decreases with $\Delta$ as the same function must explain more data. The quality of the fit does not change appreciably if $1.5 < \Delta < 4$. We take $\Delta = 2$. Importantly, as the errors in the simulated first passage problems are small compared to errors in the measured rates, the choice of $\Delta$ does not affect the comparison between theory and observation. This function is shown as the blue curve in *Figures 2 and 5b*. The confidence interval is similar to the width of the curve.

## Measurement of individual scattering numbers

We measure the scattering numbers of 31 individuals from their turning rate $\gamma$ and their swimming speed. This method is essentially identical to what is described in *Petroff et al., 2022*. These measurements are done in the same microfluidic device as the other experiments described here. At the start of the experiment, enriched consortia are directed by a 300 μT magnetic field to accumulate near a boundary of the microfluidic device, which is highlighted in red in *Figure 4*. The direction of the magnetic field is then reversed, and consortia swim into the bulk fluid. After the consortia have swum well away from the wall, the field is quickly rotated by 90° and the field intensity is reduced to $B_{\text{applied}} = 150$ μT. We used an Arduino to synchronize the microscope camera with the magnetic coils such that trajectories of consortia as they align with the new field are recorded at 21 frames/s starting from the instant that the field is switched. The individual trajectories are reconstructed using the same methods described in 'Tracking of MMB'.

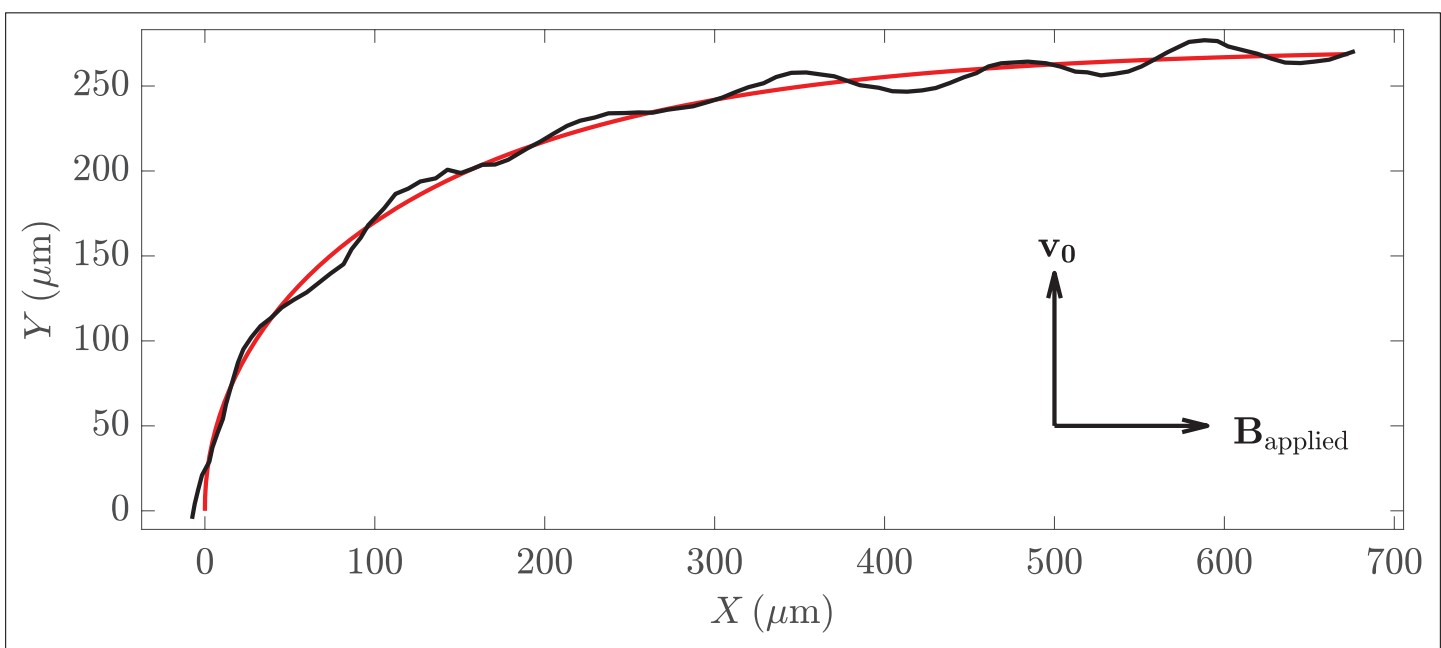

**Figure 6.** The trajectory of a consortium is shown in black. At the start of the experiment, the consortium swims with velocity $\mathbf{v}_0$ in the vertical direction and the applied magnetic field is pointed to the right. The red line shows the best fit to *Equations 6 and 7*, from which we measure $\gamma$ and $U_0$.

To extract $\gamma$ and $U_0$ from these measurements, we fit the measured trajectories of consortia to the dimensional versions of *Equations 6 and 7*. Some care must be taken in these fits. Many of the reconstructed trajectories were too short to meaningfully fit to the model. Others tracked consortia displayed escape motility (*Greenberg et al., 2005*; *Sepulchro et al., 2020*; *Yang et al., 2024*), swimming against the magnetic field. As a result of these difficulties, trajectories were selected by hand. To limit bias, two researchers independently selected trajectories that could be clearly seen turning to align with the applied field. The selections were very similar. The measured value of $\gamma$ is determined by the strength of the applied magnetic field. To find the scattering numbers, we rescale the measured value of $\gamma$ by $B_{\text{geo}}/B_{\text{applied}}$.

A trajectory of a consortium and its fit is shown in *Figure 6*. The quality of its fit is representative. Notice the sinusoidal oscillations about the predicted form. These oscillations arise if the magnetic moment of the consortium is not perfectly aligned with its velocity. They are noticeable in about half of the trajectories. As a result of these oscillations, the best fit swimming speed differs from the average instantaneous speed by about 7%. We find swimming speeds ranging from 65 μm/s to 212 μm/s.

## Acknowledgements

We would like to thank A Libchaber, A Kudrolli, A Flamholz, A Goyal, M Houssais, and O Devauchelle for their comments and assistance. E Sachinthanie measured the pore size distribution. This work was supported by the National Science Foundation (NSF PHY-2042150). Much of this reasoning was clarified at the BIRS meeting 24w5315 'Formation of Looping Networks—from Nature to Models'. We thank the organizers and the Banff International Research Station. Bill Huang at Woodruff Engineering designed the magnetic coils.

## Additional information

### Funding

| Funder | Grant reference number | Author |
|---|---|---|
| National Science Foundation | 2042150 | Alexander P Petroff |

The funders had no role in study design, data collection and interpretation, or the decision to submit the work for publication.

### Author contributions

Alexander P Petroff, Conceptualization, Data curation, Formal analysis, Supervision, Funding acquisition, Validation, Investigation, Methodology, Writing – original draft, Project administration, Writing – review and editing; Julia Hernandez, Vladislav Kelin, Nina Radchenko-Hannafin, Investigation

### Author ORCIDs

Alexander P Petroff https://orcid.org/0000-0002-9612-6593

Reviewer #1 (Public review): https://doi.org/10.7554/eLife.104797.3.sa1
Reviewer #2 (Public review): https://doi.org/10.7554/eLife.104797.3.sa2
Author response https://doi.org/10.7554/eLife.104797.3.sa3

## Additional files

### Supplementary files
MDAR checklist

## Data availability

The experimental data in figure 2 is available in *Figure 2—source data 1*, the corresponding data from the simulations is available in *Figure 2—source data 2*. The data for figure 3 are recorded in Appendix 1.

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

## Appendix 1

### Phenotypic diversity of magnetotactic bacteria

Here, we discuss the selection of species used in *Figure 3*, which includes both unicellular and multicellular species and representatives from three phyla that were enriched from four continents. *Appendix 1—figure 1* provides a key to this figure. Because the scattering number represents the ratio of the distance a magnetotactic bacteria swims as it aligns with the ambient magnetic field to the pore size, we only include papers that directly measure and report the swimming speed $U_o$ and either $U_0/\gamma$ or $\gamma$. This choice produces systematic errors in the estimates of the magnetic moments of the bacteria. Inverting measured values of $\gamma$ for the magnetic moment requires a measurement of $\mu_{\mathrm{rot}}$, which depends on the shape of the magnetotactic bacteria and the distribution of flagella on its surface. Generally, only a crude estimate of $\mu_{\mathrm{rot}}$ can be made. To limit the effect of these errors, wherever possible, we rescale the raw data that authors report.

Two classes of experiments met these criteria. The first class (*Petersen et al., 1989*) analyzes the motion of magnetotactic bacteria in a rotating magnetic field of magnitude $B_0$. If the field rotates slowly, magnetotactic bacteria swim in circular paths. At a critical period of rotation, the field rotates faster than magnetotactic bacteria can turn, and their trajectories begin to drift. The critical period $T_c = 2\pi/\gamma$. Our reported value of $\gamma_{\mathrm{geo}} = 2\pi B_{\mathrm{geo}}/T_c B_0$. The ratio of the diameter of the circular trajectory to the period of rotation gives $U_0$. The second class of experiments (*Frankel, 1984*; *Esquivel and De Barros, 1986*) analyzes the trajectory of a magnetotactic bacteria after a field reversal. The magnetotactic bacteria make a 'U-turn' of width $W = \pi U_0/\gamma$. For this class of experiments, our reported values of $\gamma_{\mathrm{geo}} = \pi U_0 B_{\mathrm{geo}}/W B_0$.

Additionally, to limit the effect of domestication on the phenotypes, the data presented is heavily biased toward experiments that extracted magnetotactic bacteria directly from the sediment, rather than relying on pure cultures. The reported values of $B_{\mathrm{geo}}$ are found by looking up the magnetic field (*NOAA, 2024*) as close as possible to the sampling location.

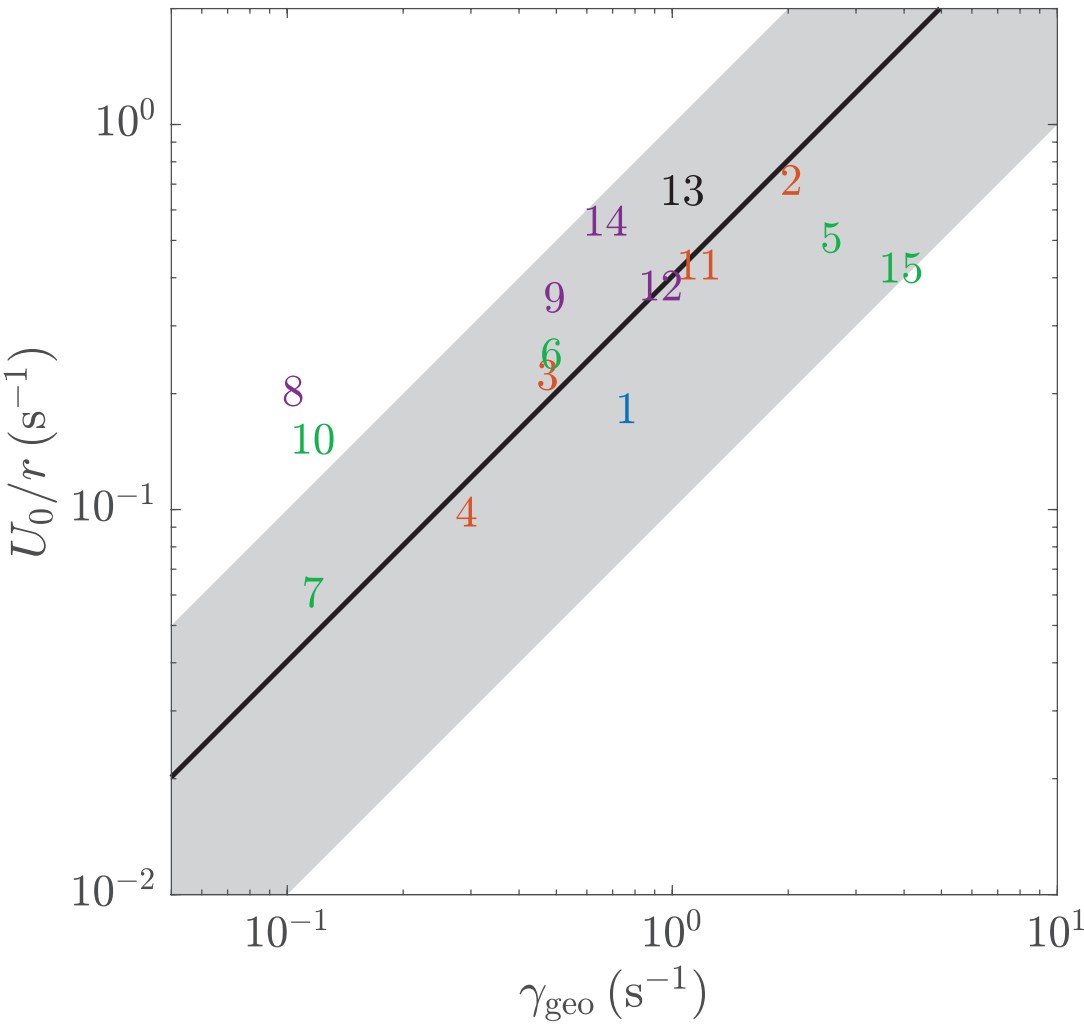

**Appendix 1—figure 1.** This companion figure to *Figure 3* provides references for each data point. Each number corresponds to the index on *Appendix 1—table 1*.

**Appendix 1—table 1.** The unidentified magnetotactic bacteria labeled MTB-(some integer) correspond to the organisms listed in Table 1 of *Esquivel and De Barros, 1986*.

| Index | Species | $B_{geo}$ ($\mu$T) | $U_0$ ($\mu$m/s) | Size($\mu$m) | $m(fAm^2)$ | $\mu_{rot}10^{18}$(Nms$^{-1}$) | $\gamma_{geo}(s^{-1})$ | Sc | Ref. |
|---|---|---|---|---|---|---|---|---|---|
| 1 | *M. bavaricum* | 49 | 36 | 9×3 | 14.9 | 1.0 | 0.8 | 0.2 | *Petersen et al., 1989; Spring et al., 1993* |
| 2 | MYC-1 | 54 | 141 | 1.7 | 1.8 | 20.9 | 2.0 | 0.3 | *Pan et al., 2009* |
| 3 | *M. magneticum AMB-1* | 47 | 44 | 3×0.9 | 0.4 | 23.5 | 0.5 | 0.5 | *Nadkarni et al., 2013* |
| 4 | *M. magneticum* | 51 | 19 | 4.7×0.7 | 0.6 | 9.4 | 0.3 | 0.3 | *Bahaj et al., 1996* |
| 5 | MTB-1 | 23 | 100 | 1 | 0.3 | 375.3 | 2.6 | 0.2 | *Esquivel and De Barros, 1986* |
| 6 | MTB-2 | 23 | 50 | 2 | 0.5 | 42.2 | 0.5 | 0.5 | *Esquivel and De Barros, 1986* |
| 7 | MTB-3 | 23 | 12 | 5×3 | 1.0 | 5.1 | 0.1 | 0.5 | *Esquivel and De Barros, 1986* |
| 8 | MTB-5 | 23 | 40 | 5 | 2.4 | 1.9 | 0.1 | 1.9 | *Esquivel and De Barros, 1986* |
| 9 | MTB-7 | 23 | 70 | 5 | 8.0 | 2.7 | 0.5 | 0.7 | *Esquivel and De Barros, 1986* |
| 10 | MTB-8 | 23 | 30 | 18×10 | 54.0 | 0.1 | 0.1 | 1.3 | *Esquivel and De Barros, 1986* |
| 11 | *Magnetococcus* | 23 | 85 | 1.3 | 8.2 | 6.2 | 1.2 | 0.4 | *Acosta-Avalos et al., 2019* |

*Appendix 1—table 1 Continued on next page*

*Appendix 1—table 1 Continued*

| Index | Species | $B_{\mathrm{geo}}$ ($\mu$T) | $U_0$ ($\mu$m/s) | Size($\mu$m) | $m(fAm^2)$ | $\mu_{\mathrm{rot}}10^{18}$(Nms$^{-1}$) | $\gamma_{\mathrm{geo}}(s^{-1})$ | Sc | Ref. |
|---|---|---|---|---|---|---|---|---|---|
| 12 | *Magnetoglobus* | 51 | 75 | 5 | 7.4 | 2.5 | 0.9 | 0.4 | *Petroff et al., 2022* |
| 13 | *Magnetoglobus* | 51 | 133 | 5 | 8.3 | 2.5 | 1.1 | 0.6 | This study |
| 14 | *Magnetoglobus* | 23 | 110 | 5.7 | 14.2 | 2.1 | 0.7 | 0.8 | *Carvalho et al., 2021* |
| 15 | Uncultured MTB coccus | 23 | 84 | 1.5 | 1.4 | 122.0 | 3.9 | 0.1 | *Carvalho et al., 2021* |

## Appendix 2

### Relationship of the transition rates and the drift velocity

Here, we derive *Equation 3*, which relates the rates $k_\pm$ at which consortia transition between pores to average speed $U_{\text{drift}}$ at which consortia move through the pore space. This relationship is useful as the transition rates can be directly measured and calculated; however, it is the drift velocity that is ecologically relevant.

Consider an infinite linear network of pores that are labeled $i = 1, 2, 3, \ldots$. Consortia swim from pore $i$, in the direction of the magnetic field, to pore $i + 1$ at a rate $k_+$. Consortia move against the magnetic field to pore $i - 1$ at a rate $k_-$. The number $N_i(t)$ of consortia in pore $i$ at time $t$ evolves in time as

$$\frac{\mathrm{d}N_i}{\mathrm{d}t} = k_+ N_{i-1} + k_- N_{i+1} - (k_+ + k_-) N_i. \tag{8}$$

This equation can be trivially rewritten as

$$\frac{\mathrm{d}N_i}{\mathrm{d}t} = \frac{(k_+ + k_-)\epsilon^2}{2} \left( \frac{N_{i+1} - 2N_i + N_{i-1}}{\epsilon^2} \right) - \epsilon \left( k_+ - k_- \right) \frac{N_{i+1} - N_{i-1}}{2\epsilon}, \tag{9}$$

where $\epsilon$ is the distance between the centers of neighboring pores. In the limit of small $\epsilon$,

$$\frac{\partial N}{\partial t} = D_{\text{eff}} \frac{\partial^2 N}{\partial x^2} - U_{\text{drift}} \frac{\partial N}{\partial x}, \tag{10}$$

where the position of the $i$th pore is $x = \epsilon i$. The effective diffusion coefficient $D_{\text{eff}} = (k_+ + k_-)\epsilon^2/2$. The drift velocity is $U_{\text{drift}} = \epsilon(k_+ - k_-)$. The distance between the centers of the pores $\epsilon \propto 2r$. The proportionality constant is determined by the ratio of the lattice spacing to the pore radius. In the square lattices analyzed here, the prefactor is 0.8344.

## Appendix 3

### Estimate of $Sc_c$ in natural pore spaces

In a random packing of grains, a typical pore is connected to other pores on all its sides of it. A simple geometric argument suggests that the critical value of $Sc_c$ in natural sediment is similar to the value found in the experiments presented here. Consider the distance $\delta$ (see *Figure 2a*) that a magnetotactic bacterium must swim to find an escape from a spherical pore with $\nu$ connections to neighboring pores. If connections are randomly and independently distributed over the boundary, $\langle \delta/r \rangle = 2/(1 + \nu)$. Thus, the pore fraction that an MMB must explore before it escapes varies only from ~0.4 to ~0.2 as the coordination number of pores varies from 4 (local tetrahedral grain packing) to 8 (local octahedral grain packing) and remains similar to the value of $\approx 0.29$ that we prescribe in our experiments. We conclude that $0.1 \lessapprox Sc_c \lessapprox 1$ and likely differs little from the critical value $Sc_c \approx 0.40$ in the experiments.

## Appendix 4

### Relationship of scattering number and body size

*Figure 3* shows that the rate $U_0/r$ that magnetotactic bacteria collide with pore boundaries is proportional to the rate $\gamma_{geo}$ that they align with the geomagnetic field. It seems plausible that this correlation reflects an anatomical relationship that is unrelated to magnetotaxis and obstacle avoidance. Perhaps larger organisms swim more quickly and produce larger magnetic moments. However, this null hypothesis can be quickly rejected.

Consider first the relationship between $\gamma_{geo} = \mu_{rot} m B_{geo}$ and body size $a$. Recall that the rotational hydraulic mobility $\mu_{rot} \propto 1/\eta a^3$, where $\eta$ is the viscosity of water, and the proportionality is determined by the shape of the organism. Thus, $\gamma_{geo} \propto m_0 c B_{geo}/\eta$, where $m_0$ is the magnetic moment of a single magnetosome and $c$ is the concentration of magnetosomes. It follows that $\gamma_{geo}$ is an intensive quantity and does not scale with $a$. From the data collected in *Appendix 1—table 1*, we find that the magnetic moment is uncorrelated with body size.

The swimming speed does not scale simply with the body size. Balancing the forces exerted by flagella with drag on the organism yields a swimming speed $U_0 \propto f_0 \sigma a/\eta$, where $f_0$ is the force exerted by a single flagellum and $\sigma$ is the surface density of flagella. While the swimming speed does scale with $a$, the variability in $\sigma$ and shape overwhelms any correlation. For example, consider MTB-2 and MTB-8 from *Appendix 1—table 1*. These organisms swim at similar speeds despite differing in size by almost an order of magnitude. Moreover, it is the smaller organism that swims faster. A comparison of all of the organisms in this table shows a weak negative correlation (–0.57) between size and swimming speed that is of marginal significance (p-value of 0.04). A comparison of 87 non-magnetotactic bacteria undertaken by *Velho Rodrigues et al., 2021*, reveals no simple relationship between swimming speed and size, suggesting that there is no biomechanical reason for $U_0$ to scale with body size.

We conclude that neither the scattering rate nor the alignment rate of bacteria scale with the body size. Rather, natural selection adapts the scattering number of a species primarily by acting upon its shape, concentration of magnetosomes, and surface density of flagella.

