## [Editor Report · eLife Assessment]

Combining experiments in microfluidic devices and computer simulation, this study provides a **valuable** analysis of the relevant parameters that determine the motility of (multicellular) magnetotactic bacteria in sediment-like environments. The study presents **convincing** evidence that there is an optimum in the biological parameters for motile life under such conditions.

---

## [Referee Report · Reviewer #1 (Public review)]

Summary:

The authors track the motion of multiple consortia of Multicellular Magnetotactic Bacteria moving through an artificial network of pores and report a discovery of a simple strategy for such consortia to move fast through the network: an optimum drift speed is attained for consortia that swim a distance comparable to the pore size in the time it takes to align the with an external magnetic field. The authors rationalize their observations using dimensional analysis and numerical simulations. Finally, they argue that the proposed strategy could generalize to other species by demonstrating the positive correlation between the swimming speed and alignment time based on theoretical analysis and parameters derived from literature.

Strengths:

The underlying dimensional analysis and model convincingly rationalize the experimental observation of an optimal drift velocity: the optimum balances the competition between the trapping in pores at large magnetic fields and random pore exploration for weak magnetic fields.

Weaknesses:

The convex pore geometry studied here creates convex traps for cells, which I expect enhances their trapping. Natural environments may create a much smaller concentration of such traps. In this case, whether a non-monotonic dependence of the drift velocity on the Scattering number would persist is unclear.

Comments on revisions:

Thank you very much for addressing my comments. I think the revisions have improved the paper.

---

## [Referee Report · Reviewer #2 (Public review)]

Summary:

The authors have made microfluidic arrays of pores and obstacles with a complex shape and studied the swimming of multicellular magnetotactic bacteria through this system. They provide a comprehensive discussion of the relevant parameters of this system and identify one dimensionless parameter, which they call the scattering number and which depends on the swimming speed and magnetic moment of the bacteria as well as the magnetic field and the size of the pores, as the most relevant. They measure the effective speed through the array of pores and obstacles as a function of that parameter, both in their microfluidic experiments and in simulations, with good agreement between the two. They find an optimal scattering number, which they estimate to reflect the parameters of the studied multicellular bacteria in their natural environment. They finally use this knowledge to compare different species. Despite the variability of bacteria parameters, they estimate the scattering number to be rather narrowly distributed, suggesting that their results apply to a broad range of species.

Strengths:

This is a beautiful experimental approach and the observation of an optimal scattering number (likely reflecting an optimal magnetic moment) is very convincing. The results here improve on similar previous work in two respects: On the one hand, the tracking of bacteria does not have the limitations of previous work, and on the other hand, the effective motility is quantified. Both features are enabled by choices of the experimental system: the use the multicellular bacteria which are larger than the usual single-celled magnetotactic bacteria and the design of the obstacle array which allows the quantification of transition rates due to the regular organization as well as the controlled release of bacteria into this array through a clever mechanism.

Weaknesses:

Some of the key experimental choices on which the strength of the approach is based also come at a price and impose some limitations, namely the use of a non-culturable organism and the regular, somewhat unrealistic artificial obstacle array, but the advantages of these choices outweigh the drawbacks.

Comments on revisions:

The paper has been improved with respect to presentation and content. In particular, I appreciate the new plots comparing the simulation and experiments directly and the estimate of the scattering number for different species. In my opinion, all issues raised by the reviewers have been addressed in a productive way.

---

## [Author Response]

The following is the authors’ response to the original reviews

**Public Reviews:**

**Reviewer #1 (Public review):**
The authors track the motion of multiple consortia of Multicellular Magnetotactic Bacteria moving through an artificial network of pores and report a discovery of a simple strategy for such consortia to move fast through the network: an optimum drift speed is attained for consortia that swim a distance comparable to the pore size in the time it takes to align the with an external magnetic field. The authors rationalize their observations using dimensional analysis and numerical simulations. Finally, they argue that the proposed strategy could generalize to other species by demonstrating the positive correlation between the swimming speed and alignment time based on parameters derived from literature.Strengths:The underlying dimensional analysis and model convincingly rationalize the experimental observation of an optimal drift velocity: the optimum balances the competition between the trapping in pores at large magnetic fields and random pore exploration for weak magnetic fields.Weaknesses:The convex pore geometry studied here creates convex traps for cells, which I expect enhances their trapping. The more natural concave geometries, resulting from random packing of spheres, would create no such traps. In this case, whether a non-monotonic dependence of the drift velocity on the Scattering number would persist is unclear.

We agree that convex walls increase the time that consortia remain trapped in pores at high magnetic fields. Since the non-monotonic behavior of the drift velocity with the Scattering number arises largely due to these long trapping times, we agree that experiments using concave pores are likely to show a peak drift velocity that is diminished or erased.

However, we disagree that a random packing of spheres or similar particles provides an appropriate model for natural sediment, which is not composed exclusively of hard particles in a pure fluid. Pore geometry is also influenced by clogging. Biofilms growing within a network of convex pillars in two-dimensional microfluidic devices have been observed to connect neighboring pillars, thereby forming convex pores. Similar pore structures appear in simulations of biofilm growth between spherical particles in three dimensions. Moreover, the salt marsh sediment in which MMB live is more complex than simple sand grains, as cohesive organic particles are abundant. Experiments in microfluidic channels show that cohesive particles clog narrow passageways and form pores similar to those analyzed here. Thus, we expect convex pores to be present and even common in natural sediment where clogging plays a role.

The concentration of convex pores in the experiments presented here is almost certainly much higher than in nature. Nonetheless, since magnetotactic bacteria continuously swim through the pore space, they are likely to regularly encounter such convexities. Efficient navigation of the pore space thus requires that magnetotactic bacteria be able to escape these traps. In the original version of this manuscript, this reasoning was reduced to only one or two sentences. That was a mistake, and we thank the reviewer for prompting us to expand on this point. As the reviewer notes, this reasoning is central to the analysis and should have been featured more prominently. In the final version, we will devote considerable space to this hypothesis and provide references to support the claims made above.

The reviewer suggests that the generality of this work depends on our finding a ”positive correlation between the swimming speed and alignment [rate] based on parameters derived from literature.” We wish to emphasize that, in addition to predicting this correlation, our theory also predicts the function that describes it. The black line in Figure 3 is not fitted to the parameters found in the literature review; it is a pure prediction.

**Reviewer #2 (Public review):**
The authors have made microfluidic arrays of pores and obstacles with a complex shape and studied the swimming of multicellular magnetotactic bacteria through this system. They provide a comprehensive discussion of the relevant parameters of this system and identify one dimensionless parameter, which they call the scattering number and which depends on the swimming speed and magnetic moment of the bacteria as well as the magnetic field and the size of the pores, as the most relevant. They measure the effective speed through the array of pores and obstacles as a function of that parameter, both in their microfluidic experiments and in simulations, and find an optimal scattering number, which they estimate to reflect the parameters of the studied multicellular bacteria in their natural environment. They finally use this knowledge to compare different species to test the generality of this idea.Strengths:This is a beautiful experimental approach and the observation of an optimal scattering number (likely reflecting an optimal magnetic moment) is very convincing. The results here improve on similar previous work in two respects: On the one hand, the tracking of bacteria does not have the limitations of previous work, and on the other hand, the effective motility is quantified. Both features are enabled by choices of the experimental system: the use the multicellular bacteria which are larger than the usual single-celled magnetotactic bacteria and the design of the obstacle array which allows the quantification of transition rates due to the regular organization as well as the controlled release of bacteria into this array through a clever mechanism.Weaknesses:Some of the reported results are not as new as the authors suggest, specifically trapping by obstacles and the detrimental effect of a strong magnetic field have been reported before as has the hypothesis that the magnetic moment may be optimized for swimming in a sediment environment where there is a competition of directed swimming and trapping. Other than that, some of the key experimental choices on which the strength of the approach is based also come at a price and impose some limitations, namely the use of a non-culturable organism and the regular, somewhat unrealistic artificial obstacle array.

In the “Recommendations for the Authors,” this reviewer drew our attention to a manuscript that absolutely should have been prominently cited. As the reviewer notes, our manuscript meaningfully expands upon this work. We are pleased to learn that the phenomena discussed here are more general than we initially understood. It was an oversight not to have found this paper earlier. The final version will better contextualize our work and give due credit to the authors. We sincerely appreciate the reviewer for bringing this work to our attention.

We disagree that the use of non-culturable organisms and our unrealistic array should be considered serious weaknesses. While any methodological choice comes with trade-offs, we believe these choices best advance our aims. First, the goal of our research, both within and beyond this manuscript, is to understand the phenotypes of magnetotactic bacteria in nature. While using pure cultures enables many useful techniques, phenotypic traits may drift as strains undergo domestication. We therefore prioritize studying environmental enrichments.

Clearly, an array of obstacles does not fully represent natural heterogeneity. However, using regular pore shapes allows us to average over enough consortium-wall collisions to enable a parameter-free comparison between theory and experiment. Conducting an analysis like this with randomly arranged obstacles would require averaging over an ensemble of random environments, which is practically challenging given the experimental constraints. Since we find good agreement between theory and experiment in simple geometries, we are now in a position to justify extending our theory to more realistic geometries. Additionally, we note that a microfluidic device composed of a random arrangement of obstacles would also be a poor representation of environmental heterogeneity, as pore shape and network topology differ between two and three dimensions.

**Recommendations for the Authors:**

**Reviewer #1 (Recommendations for the authors):**
My main suggestion is for the authors to describe the limitations of their approach in the case of concave pores.As we noted in our public comments, this was a very useful comment to hear from you and one that has been repeated as we have spoken about these results to colleagues. Convexities here represent an experimentally simple way to force bacteria to back track through the maze, as they must through natural sediment. We have greatly expanded this discussion to clarify this reasoning (lines 84–105). We provide references to three types of physical processes that may lead to such traps. First, as in figure 1 of Kurz et al, biofilm (white) can fill the spaces between convex pillars to create covexities. Additionally, clogging by cohesive particles can make narrow passageways between convex particles impassible. An example of clogging is shown in figure 6 of Dressaire & Sauret 2017. Finally, air bubbles trapped in the sediment can create pore-scale dead ends that require bacteria to backtrack. The full references are provided in the main text.Small points:(1) How many trajectories were used to produce Figures 2 b and c?

We have modified the caption to note that these data represent the measured transition rates of a total 938 consortia at various Scattering numbers. Each consortium may pass between pores many times.

(2) Can the authors describe in more detail how Equation (3) is derived? Why doesn’t it depend on the gap size between the pores?

We have provided a derivation of this equation in Appendix 2 of the new version. This derivation shows that the drift velocity *U*_drift_ is proportional to the pore diameter and difference between the transition rates.

The proportionality constant *α* depends on how the pores are connected together in space. In the original version, we wanted to highlight the role of the asymmetry of the transition rates, so we imagined a one dimensional network of pores without gaps. In this case, *α* = 1. This reasoning was poorly explained in the previous version and we thank the reviewer for pointing this deficiency out. In the new version, we include the gap size and use the layout of pores in a square lattice with gaps, which is shown in figure 1. The proportionality constant for a square lattice in the absence of gaps√ would be 1/2. The limitations of photolithography require some gap that increase the proportionality constant to *α* = 0.8344.

We have updated the text, equation (3), and the figures to account for the finite gap sizes.

(3) I found the second part of the abstract, related to the comparison between diverse bacteria, to be slightly misleading. Upon first reading, my expectation was that the authors carried out experiments with different species.

We have modified the abstract to make clear that we rely on values taken from a literature review.

(4) More information is needed on how many trajectories were used to produce the probability densities in Figures 1b-d. How were the densities computed?

The probability distributions give the probability that a pixel in a pore is covered by a consortium. They reflect between 1.2 and 7 million measurements (depending on the panel) of the instantaneous positions of consortia. We have added a section (Lines 453–469) to Materials and Methods that describes exactly how these distributions were calculated.

**Reviewer #2 (Recommendations for the authors):**
(1) As mentioned under Weaknesses in the Public review, some results are less new than claimed here. The existence of an optimal magnetic moment has been shown by Codutti et al eLife eLife13:RP98001 in very similar experiments, where it was also proposed that this may be an evolutionary adaptation to the sediment habitat. The paper here provides additional evidence for this, and with better tracking and quantification, but previous work should be discussed. Likewise, the work by Dekharghani et al. that is mentioned rather suddenly in the Results section appears to be a crucial previous state of the art and could already be mentioned in the introduction.

We thank the reviewer for bringing this paper, which came out as we were writing this manuscript, to our attention. The hypothesis that there is an optimal phenotype that balances magnetotaxis with obstacle avoidance—and that natural selection could guide organisms to this optimum—goes back to at least 2022. It seems that Codutti et al independently came up with this same hypothesis and provided the first test.

We have substantively rewritten the introduction (Lines 46–58) to better contextualize our work and give due attention to Dekharghani et al.

(2) The first paragraph of Results also contains background information and could be moved into the introduction.

As part of the rewrite to better contextualize our work, we moved the first two paragraphs of results to the introduction.

(3) I found Figure 1 a bit confusing and it took me some time to understand the geometry. I think the black obstacles are very dominant to the viewer’s eye and draw attention away from the essentially circular shape of the pores. Likewise, I am not sure that cutting the neighboring pores off in a circular fashion in Figures 1b-d was the best choice. The authors should think about whether the presentation can be improved. Likewise, when describing the direction of the field in the text, I would suggest adding that it is along the horizontal direction in Figure 1.

We have modified the figure and the text as the reviewer suggests.

(4) That collisions with a pore wall are an important mechanism of changing direction is clear and it is nice to see the paper demonstrate that this mechanism is dominant over rotational diffusion. However, this may not be universal, as (i) rotational diffusion is more important for smaller cells and (ii) interaction with walls can result in all kinds of different behaviors than complete randomization (e.g. swimming along the walls as shown in microfluidic chambers, Ostapenko et al. Phys Rev Lett 2018, Codutti et al. eLife 2022, or reversals, Kuhn et al PNAS 2017). Here, it appears that complete randomization of the direction is an assumption, but this could be tested/quantified by analyzing the trajectories.

This is an excellent point. We have modified the text to describe qualitatively how these tendencies would shift the Critical Scattering number. We also note in the text that there is evidence of these differences in Fig 3. The Desulfobacterota are shifted upwards in Fig 3 relative to the *α*-proteobacteria. This shift indicates that Desulfobacterota tend to live at slightly greater scattering numbers of 0.9±0.3 than the α-proteobacteria, which live at scattering number 0.37 ± 0.03. It is likely that this difference reflects taxonomic differences in rotational diffusion and cell-wall interactions.

It is true that total randomization of the direction is indeed an assumption, and it is stated as such in line 189. We performed all of the numerics to find the solid curves in Fig 2 before we got any experimental data and so, at the time, total randomization seemed like a fair choice. Looking at Fig 2b, it is clear that these numerics systematically overestimate *k*_−_. We believe that this error is do to the assumption of total randomization.

As this effect is small and does not change any of the conclusions of the paper and Codutti *et al* were able to publish their paper in the time that we were writing ours, we feel some urgency to move forward.

(5) From the manuscript it is not fully clear to what extent experiments and simulations are or can be quantitatively compared. For example: is the curve (“fit”) in Figure 2c based on the simulations? Is there an explicit expression or is this just a spline or something like that? Why does Figure 5 (simulation) show the velocity as a function of Sc^−1^and Figure 2 (experiment) as a function of Sc? It looks to me as if a quantitative comparison could be achieved.

The original version of Figure 2 shows a quantitative comparison between theory and experiment with no fit parameters. The data points are the result of experiments in which consortia are tracked as they as they move between connected pores. The solid line is a found by interpolating a smooth curve through the data from simulations. As we make clear in the new version (Lines 537–551), this blue curve is the most probable smooth curve that explains the simulations.

We have added the simulations to figure 2 so that a single panel includes the data, the simulations, and the smooth curve. To further make clear that this comparison is quantitative and parameter free, we have added a panel to Figure 2. This panel directly compares the prediction to observation and is independent of the blue curve.

As was noted (deep within the methods section) in the original version, our numerics can exactly simulate Sc = ∞. Consequently, it was reasonable to simulate parameters that are uniformly spaced in Sc^−1^.

(6) While I like the idea behind Figure 3, the data shown here is not as convincing as suggested. If one looks at the data without the black line, I think one gets a weaker dependence. The correlation between *U*_0_ and *γ*_geo_ is likely not as strong as it seems. Calculating a correlation coefficient might be helpful here. In any case, the assumptions going into this figure should be discussed more explicitly and the results should in my opinion be phrased more cautiously (I tend to believe what the authors claim, but I don’t think the evidence for this point is very strong).

We appreciate the reviewer’s skepticism. However, we believe that the data are stronger than one might understand from the previous text. We have rewritten the text (Lines 219–291) and included new analysis, figures, and explanation to make three points clear.

(a) It is surprising that speed, magnetic moment, and mobility all vary tremendously(between one and three orders of magnitude) across taxa and environment, however, their dimensionless combination Sc is narrowly distributed. We have added a panel to Fig. 3 to show the measured Scattering numbers.

It is notable that there are no adjusted parameters in the calculation of the Scattering numbers: it is a simple dimensionless combination of phenotypic and environmental parameters. All but one of these parameters (the pore size) is measured either by us or by other authors. The pore radius is likely narrowly distributed. We measure it at our field site and, when it is not reported, we use a value typical of the geological and fluvial environment. Just as the size of sand grains does not vary greatly between the beaches of Australia, Africa, and California, it is a good assumption that the pore spaces that host these magnetotactic bacteria do not vary tremendously in size.

(b) In the new version we compare the Scattering number statistics to a parameterfree null model of phenotypic diversity. We argue in the text that it is appropriate to bootstrap over the phenotypic diversity of species. This null model provides the correct method to calculate *p*-values as the variability in the Scattering numbers is neither identically distributed nor normally distributed.

We use this null model to show that—given the measured phenotypic diversity across species—the probability that fifteen random species would fall within the measured range of Scattering numbers that is consistent with optimal navigation is ∼ 10^−6^. This result is strong evidence that the phenotypic variables exhibit the correlations that are predicted by our analysis.

(c) The correlation between *U0/r* and *γ*_geo_ is reasonably strong. I think that our choice of axes in Fig 3, which were chosen to fit the legend, make the data look flatter than then they actually are. Here are the same data plotted without the line with tighter axes:

With the exception of the very first point and the very last point, the data appear to our eyes to be pretty correlated. This impression is born out by a calculation of the correlation coefficient which gives 0.77. The *p*-value is 4 × 10^−4^. We have included these values in the main text to clarify that this correlation is both statistically significant and of primary importance.

(7) There is a comment at the end of the discussion that the evolutionary hypothesis could be tested by transferring the magnetotaxis genes to nonmagnetotactic organisms. This would indeed be highly desirable, but this is very difficult as indicated by the successful efforts in that direction (which often are only moderately magnetic/magnetotactic), see Kolinko et al Nature Nanotech 2014, Dziuba et al Nature Nanotech 2024.

Thank you for highlighting these references, which we have included. We agree that these experiments will be challenging. Our results make a prediction about the evolution of these strains, so it seems worth mentioning this fact. We feel that this manuscript is not the correct space for a detailed description of challenges that we will encounter should we pursue this direction of study.

(8) A section on how the bacterial samples were obtained could be added in Methods.

We have done so.

Additional Changes

(1) In the original version, we feared that the consortia in the microfluidic device arepoorly representative of the natural population. Consequently, we used the values from previous experiments, which we performed using consortia taken from the same pond. Since submitting this manuscript we have undertaken new experiments that allowed us to measure the Scattering number of individual consortia. It turns out the effect is smaller than we worried. We have included these measurements in the new version. We find that even as the most common phenotypes vary over the course of time, the Scattering number remains constant. This result is additional evidence that there is strong selective pressure to optimally navigate.

As a result of these additions, we have added an author, Julia Hernandez, who contributed to these experiments and analysis.

(2) We have expanded the table of phenotypic variable in Appendix 1 to make it easier forother researchers to reproduce our analysis.